Letter

# Primate-specific ZNF808 is essential for pancreatic development in humans

Elisa De Franco [1,29], Nick D. L. Owens [1,29], Hossam Montaser[2,29], Matthew N. Wakeling [1], Jonna Saarimäki-Vire[2], Athina Triantou[3], Hazem Ibrahim [2], Diego Balboa [4,5], Richard C. Caswell [6], Rachel E. Jennings[7,8], Jouni A. Kvist [2], Matthew B. Johnson[1], Sachin Muralidharan[2], Sian Ellard[6], Caroline F. Wright [1], Sateesh Maddirevula[9], Fowzan S. Alkuraya [9,10], Pancreatic Agenesis Gene Discovery Consortium*, Neil A. Hanley [7,8], Sarah E. Flanagan [1], Timo Otonkoski [2,11,30] ✉, Andrew T. Hattersley [1,30] ✉ & Michael Imbeault [3,30] ✉

Identifying genes linked to extreme phenotypes in humans has the potential to highlight biological processes not shared with all other mammals. Here, we report the identification of homozygous loss-of-function variants in the primate-specific gene *ZNF808* as a cause of pancreatic agenesis. ZNF808 is a member of the KRAB zinc finger protein family, a large and rapidly evolving group of epigenetic silencers which target transposable elements. We show that loss of ZNF808 in vitro results in aberrant activation of regulatory potential contained in the primate-specific transposable elements it represses during early pancreas development. This leads to inappropriate specification of cell fate with induction of genes associated with liver identity. Our results highlight the essential role of *ZNF808* in pancreatic development in humans and the contribution of primate-specific regions of the human genome to congenital developmental disease.

Studying the genetic basis of congenital diseases in humans can reveal unique mechanisms orchestrating human organ development that are distinct from those in other species. The development of the human pancreas has unique characteristics compared to its counterpart in rodents, underscoring the importance of investigating this process in humans to elucidate species-specific regulation mechanisms. Previous human genetic studies of individuals with pancreatic agenesis, a rare congenital condition resulting from inappropriate pancreas development, identified evolutionarily conserved genes (*GATA6* (ref. 1), *GATA4* (ref. 2) and *CNOT1* (ref. 3)) involved in the early stages of pancreatic development which have different dosage-dependent effects in mouse and human. Here, we report the identification of loss-of-function variants in the primate-specific gene *ZNF808* as a genetic cause of defective pancreatic development in humans.

[1]Institute of Clinical and Biomedical Sciences, University of Exeter Faculty of Health and Life Sciences, Exeter, UK. [2]Stem Cells and Metabolism Research Program, Faculty of Medicine, University of Helsinki, Helsinki, Finland. [3]Department of Genetics, University of Cambridge, Cambridge, UK. [4]Regulatory Genomics and Diabetes, Centre for Genomic Regulation, Barcelona Institute of Science and Technology, Barcelona, Spain. [5]Centro de Investigación Biomédica en Red de Diabetes y Enfermedades Metabólicas Asociadas (CIBERDEM), Barcelona, Spain. [6]Genomics Laboratory, Royal Devon University Healthcare NHS Foundation Trust, Exeter, UK. [7]Division of Diabetes, Endocrinology & Gastroenterology, Faculty of Biology, Medicine & Health, University of Manchester, Manchester, UK. [8]Endocrinology Department, Manchester University NHS Foundation Trust, Manchester, UK. [9]Department of Translational Genomics, Center for Genomic Medicine, King Faisal Specialist Hospital and Research Center, Riyadh, Saudi Arabia. [10]Department of Anatomy and Cell Biology, College of Medicine, Alfaisal University, Riyadh, Saudi Arabia. [11]Children's Hospital, Helsinki University Hospital and University of Helsinki, Helsinki, Finland. [29]These authors contributed equally: E. De Franco, N. D. L. Owens, H. Montaser. [30]These authors jointly supervised this work: T. Otonkoski, A. T. Hattersley, M. Imbeault. *A list of authors and their affiliations appears at the end of the paper. ✉e-mail: timo.otonkoski@helsinki.fi; A.T.Hattersley@exeter.ac.uk; mi339@cam.ac.uk

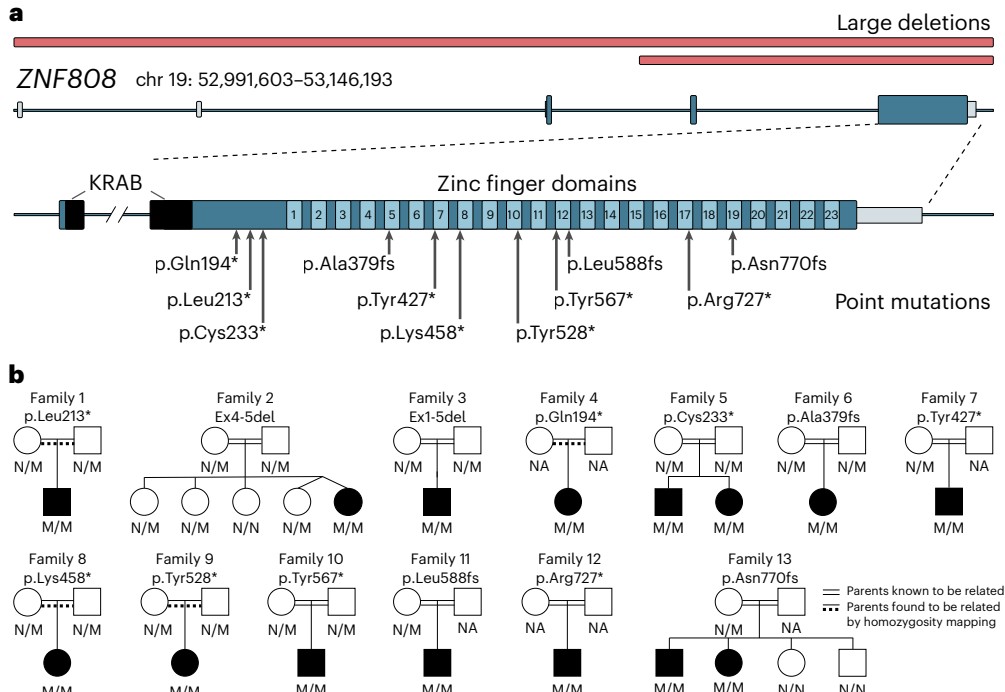

**Fig. 1 | Identification of homozygous variants in *ZNF808* as a cause of pancreatic agenesis. a**, Schematic representation of the *ZNF808* gene and the pathogenic variants identified in 13 families. Deletions are highlighted in red above the gene cartoon while the loss-of-function variants are represented below. The KRAB and zinc finger domains are annotated at the gene level. **b**, Partial pedigrees of the 13 families with homozygous *ZNF808* variants. NA, sample not available for testing; N/N, variant not detected; N/M, heterozygote for variant; M/M, homozygote for variant. Two black lines between parents indicate individuals who were known to be related. A black line and a dashed line between parents indicate individuals who were not known to be related at testing but were confirmed to be consanguineous by homozygosity mapping of next-generation sequencing data calculated with SavvyHomozygosity[34].

To identify genetic causes of defective pancreas development, we initially studied two unrelated individuals with isolated pancreatic agenesis (defined by neonatal diabetes, which is diagnosed before 6 months of age and exocrine pancreatic insufficiency[1]) in whom all known genetic causes had been excluded. We performed exome sequencing in the two affected individuals and their consanguineous parents and found that both individuals were homozygous for *ZNF808* loss-of-function variants (Supplementary Tables 1 and 2).

We next investigated the presence of rare *ZNF808* biallelic variants in 233 more patients with neonatal diabetes without a pathogenic variant in the known etiological genes[4]. Homozygous loss-of-function *ZNF808* variants were identified in a further 11 unrelated individuals and 2 affected siblings (Fig. 1a,b and Supplementary Table 3). All frameshift and stop-gain variants affect residues in the last exon of the gene and are predicted to result in a truncated protein lacking between 4 and all 23 of the zinc finger domains. Two patients (probands 2 and 3) were homozygous for deletions predicted to result in no *ZNF808* messenger RNA. All families showed cosegregation of the *ZNF808* homozygous variants with the disease consistent with recessive inheritance (Fig. 1b). No deleterious homozygous loss-of-function *ZNF808* variants were identified in >680,000 individuals without neonatal diabetes or pancreatic agenesis (UK BioBank (*n* = 454,756), Gnomad v.2.1.1 (*n* = 141,071), 100,000 genomes project from Genomics England (*n* = 75,118) and Genes and Health (*n* = 8,921)). Overall, our findings show that *ZNF808* biallelic loss-of-function variants cause pancreatic agenesis/neonatal diabetes.

The patients' phenotype supports a role for *ZNF808* in early pancreatic development affecting both endocrine and exocrine pancreatic functions (pancreatic agenesis). All individuals showed markedly reduced insulin secretion in utero, as they all had low birth weight (median −2.98 s.d., interquartile range (IQR) −3.56 to −2.47) reflecting reduced insulin-mediated fetal growth and also postdelivery, as they rapidly developed insulin-requiring diabetes (median age at diagnosis

17 weeks, IQR 3–23). There was evidence of early exocrine failure, with 5 of 5 patients tested having undetectable fecal elastase and 4 further patients having symptoms or investigations consistent with fat malabsorption (Supplementary Table 3). The patients' phenotype is restricted to pancreas as no extra-pancreatic features were consistently observed in the cohort.

Analysis of transcriptomic data from GTEx reveals that *ZNF808* is broadly expressed across human adult tissues (Extended Data Fig. 1a). Transcriptomic analysis of 15 human embryonic tissues[5] showed that *ZNF808* is maximally expressed in developing pancreas (Extended Data Fig. 1b).

ZNF808 is a member of the KRAB zinc finger protein (KZFPs) family, the largest group of DNA binding factors in the human genome (~350 protein-coding genes). KZFPs primarily act as epigenetic silencers of transposable elements through establishment of heterochromatin-associated H3K9me3 (refs. 6,7). The KZFP family is rapidly evolving with new members found at most phylogenetic branches since its emergence at the dawn of tetrapods[8,9].

*ZNF808* is exclusively found in primates and its evolutionary origin can be functionally traced through its zinc finger signature to a common ancestor of Old World monkeys (Fig. 2a), with no similar array of zinc fingers found in New World monkeys or any other mammals[8]. Our genome-wide analysis of sequence identity between primate and non-primate mammalian orthologues for all human protein-coding genes did not highlight any other primate-specific gene confirmed to be causative for a human developmental disorder (Extended Data Fig. 1c and Supplementary Table 4). These findings confirm how extremely rare it is for a primate-specific gene to cause a human congenital developmental disorder.

Genome-wide binding profiling of ZNF808 shows that it primarily targets the long terminal repeat of endogenous retroviruses classified as MER11 elements which comprise subfamilies A, B and C (Fig. 2b).

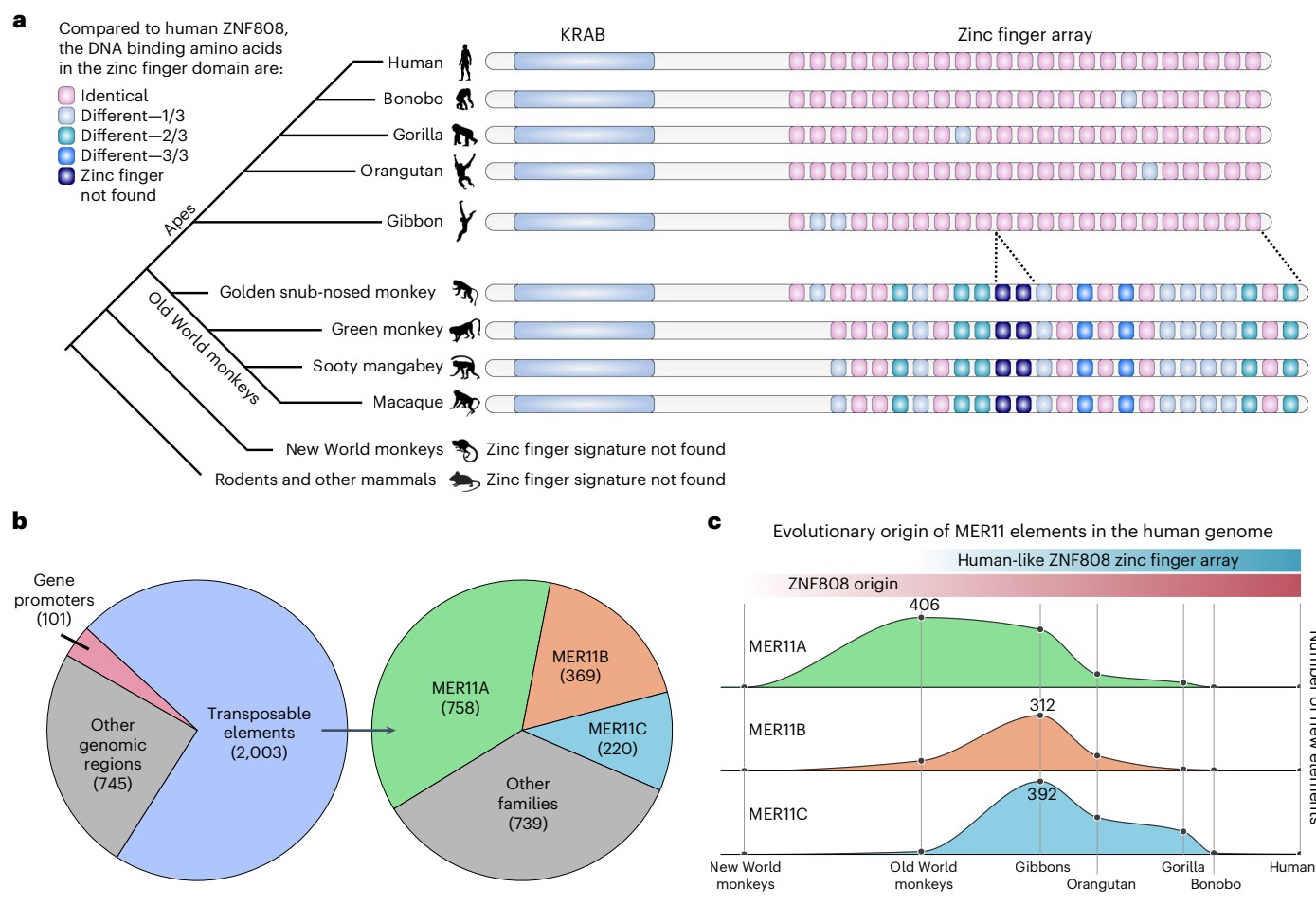

**Fig. 2 | *ZNF808* is a primate-specific gene targeting transposons of similar evolutionary age. a**, Reconstructed phylogeny of ZNF808 using a zinc finger signature approach. The three amino acids of each zinc finger directly contacting DNA were used to build a specific functional signature to track evolution of ZNF808, as previously described[8]. Zinc finger domains are color-coded according to the number of variants in each triplet compared with the human version. Notable events of loss or gain of zinc fingers are also represented. No appreciable homology with any zinc finger array was detected in New World monkeys or in any other mammals. Silhouettes of representative species are all from PhyloPic.org. **b**, ZNF808 binds primarily MER11 transposable elements. Analysis of ZNF808 ChIP–seq data[8] reveals that it primarily intersects with transposable elements, although a few binding sites are found on gene promoters and other genomic regions. Further analysis of transposons shows that ZNF808 binds primarily elements of the MER11 family—MER11A, MER11B and MER11C. **c**, MER11 transposable elements are primate-specific. The origin of each individual MER11 element in the human genome was traced using a comparative multiple alignment of 241 species[35]. The age of each element was determined as corresponding to the farthest phylogenetic branch where we could find a similar copy at a syntenic locus. Data points are plotted as the sum of elements found to have originated at each phylogenetic branch per subfamily—the *x* axis is scaled according to time in million years from estimates between the human genome and each phylogenetic branch common ancestor. A curve is interpolated between the data points to show the estimated rate of replication between phylogenetic branches. The scale is relative to each subfamily and is indicative of proportional changes between phylogenetic categories—the highest point for each subfamily is annotated with the number of new elements for scale.

These elements also originated in Old World monkeys and spread during evolution in successive waves until recently (Fig. 2c). It was suggested that remnants of MER11 elements might be domesticated[10–13], providing regulatory potential that can modulate gene expression in certain cellular contexts even though they have lost their ability to transpose. Using the chromatin immunoprecipitation ChIP-Atlas[14] database, we found that MER11 elements show enriched occupancy for many DNA binding factors, including some involved in early embryonic development (Extended Data Fig. 2a,b). Three of these (GATA4, GATA6 and HNF4A) are known causes of pancreatic agenesis and/or diabetes in humans when mutated[1,2,15,16] and are involved in endoderm, liver and pancreas specification. We also found that subsets of MER11 elements are targeted by other primate-specific KZFPs (Extended Data Fig. 2b). This suggests that MER11 elements have the potential to be regulatory elements and are silenced by a group of primate-specific KZFPs, with ZNF808 being the central effector.

To characterize the molecular events triggered by *ZNF808* loss, we functionally inactivated *ZNF808* in H1 human embryonic stem cells (hESC) using CRISPR-Cpf1 (hereafter, *ZNF808* KO) (Fig. 3a and Extended Data Fig. 3). Also, we generated induced pluripotent stem cells (iPSCs) derived from proband 2 harboring a homozygous deletion of *ZNF808* exons 4 and 5 to validate our findings.

We assayed epigenetic changes at key stages of in vitro differentiation from pluripotent stem cells to pancreatic progenitor cells (Fig. 3b)[17]. We first quantified the genome-wide presence of H3K9me3 (Fig. 3c, top), a hallmark of heterochromatin induced by KZFPs[18] and focused our analysis on MER11 elements as they represented 91.3% of ZNF808-bound sites with H3K9me3 (Extended Data Fig. 4a–c). H3K9me3 was detected on 1,367 MER11 elements in wild-type (WT) stem cells, 1,041 of which are known to be targets of ZNF808. The *ZNF808* KO cells show a loss of 693 H3K9me3-positive MER11 loci at S0 (embryonic stem cells), with 684 of these being known targets of ZNF808. The

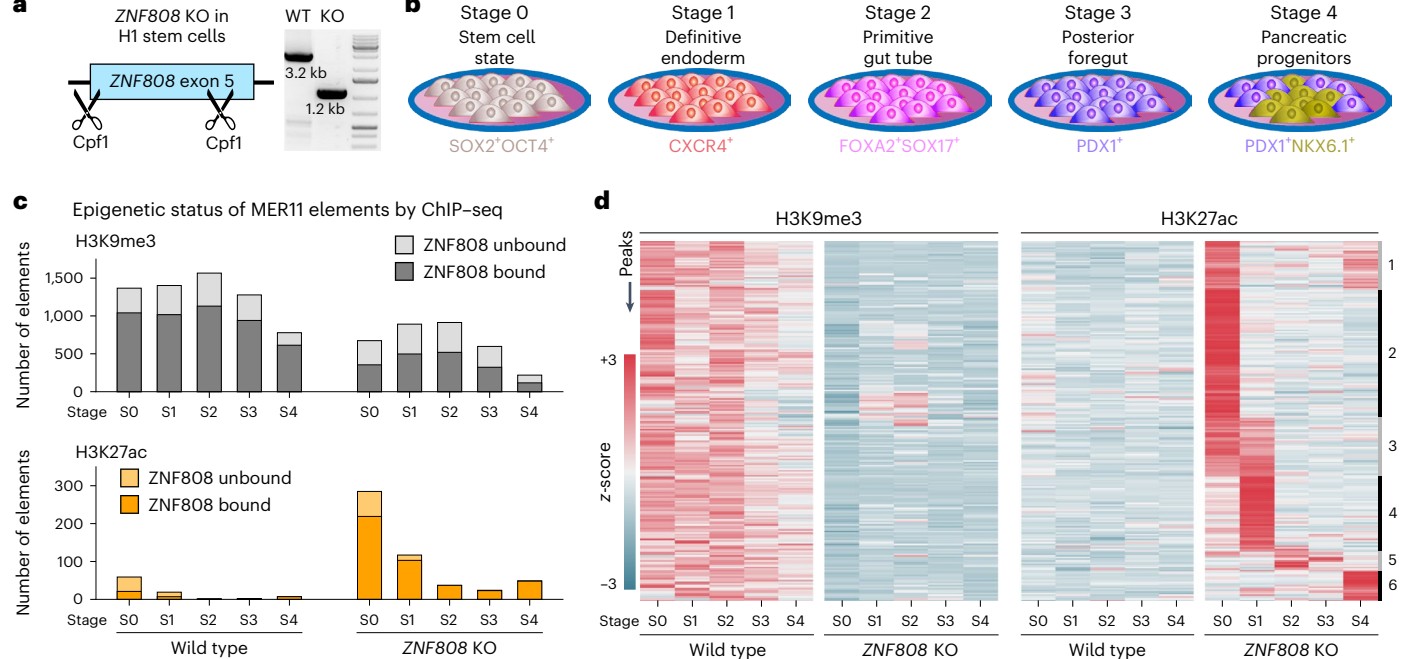

**Fig. 3 | Loss of *ZNF808* unmasks the regulatory potential of MER11 elements during pancreatic differentiation. a**, *ZNF808* gene editing using CRISPR. Cpf1 and two guide RNAs targeting the zinc finger array were used in H1 stem cells to produce the *ZNF808* KO. **b**, In vitro differentiation protocol to pancreatic progenitors. Overview of the multistep differentiation protocol used in this study for both epigenetic and transcriptomic analysis of differences induced by *ZNF808* KO. Stage numbers from S0 to S4 are used through the text to refer to specific steps. **c**, ZNF808 is important for the maintenance of epigenetic repression on MER11 elements. Left, H3K9me3 ChIP–seq peaks intersecting with MER11 elements reveal that most sites are covered in heterochromatin-associated H3K9me3 and are bound by ZNF808. A proportion of these peaks lose H3K9me3 in *ZNF808* KO clones at various stages of differentiation (top). Results show that many MER11 elements that had H3K9me3 signal in the WT gain H3K27ac in the *ZNF808* KO, especially at the early stages of differentiation (bottom). **d**, A subset of MER11 elements is activated in the *ZNF808* KO heatmap showing clustering of 220 MER11 elements displaying a loss of H3K9me3 followed by a gain of H3K27ac in at least one stage of differentiation. Normalized ChIP–seq signal is shown—color scale ranges from +3 (red) to −3 (blue) on a *z*-score scale.

loss of H3K9me3 on MER11 elements known to be bound by ZNF808 is statistically significant when considering all stages (Fisher exact test $P < 3.6 \times 10^{-61}$). This implies that ZNF808 plays a key role in silencing MER11 elements in early development.

We then surveyed whether the loss of silencing revealed active regulatory potential at MER11 elements. In the *ZNF808* KO, we observed emergence of 226 H3K27ac-positive MER11 elements at S0 (Fig. 3c, bottom), most (198) at known targets of ZNF808. The gain of H3K27ac on known MER11 elements targeted by ZNF808 is statistically significant when all stages are considered (Fisher exact test $P < 1.8 \times 10^{-16}$).

We identified a subset of 220 MER11 elements where both loss of H3K9me3 and gain of H3K27ac were observed at the same stage in the *ZNF808* KO. These have distinct patterns of activity during differentiation which we could group in six clusters (Fig. 3d and Extended Data Fig. 5), reflective of when the elements gain H3K27ac. We found that MER11 elements gaining activity in the *ZNF808* KO later during differentiation (cluster nos. 5 and 6) are more frequently bound by GATA4 or members of the HNF4 family (Extended Data Fig. 5c). These transcription factors are dynamically expressed during our differentiation protocol (Extended Data Fig. 6) and are known to be important in specification of endodermal derived tissues[19,20] including pancreas[21,22] and liver[20,23,24]. Analysis of chromatin accessibility from a single-cell ATAC-seq tissue atlas[25] and the NIH Roadmap[26] dataset shows that MER11 elements are not active in pancreas, however multiple MER11 loci are normally active in specific cell types, including fetal trophoblasts, enterocytes and hepatocytes (Extended Data Fig. 7a,d,e), with enrichment for those unmasked in the *ZNF808* KO cells (Extended Data Fig. 7b,c). We found strong agreement between our *ZNF808* KO and the patient-derived iPSCs in terms of loss of H3K9me3 and gain of H3K27ac (Extended Data Fig. 8a).

These results show that ZNF808 silences specific MER11 elements during differentiation toward pancreatic lineages and that loss of ZNF808 unmasks their regulatory potential which is normally found active in other cellular contexts.

We next quantified transcriptomic changes induced in the *ZNF808* KO during differentiation. We found that the number of genes whose expression was dysregulated increased from 443 genes at S0 (stem cell state) to a peak of 2,124 genes at S3 (posterior foregut) (Fig. 4a). Comparable changes in gene expression were detected in the patient-derived iPSCs by both RNA-seq and quantitative PCR (qPCR) with reverse transcription (Extended Data Fig. 8b,c). We found that unmasked MER11 elements drive proximal gene activation in the *ZNF808* KO predominantly at the first two stages of differentiation (S0, stem cell state; S1, definitive endoderm), with MER11 elements over-represented near activated genes at enhancer (10 kb–1 Mb) but not promoter distances (Fig. 4b and Extended Data Fig. 9a–c). We also found that the smaller group of MER11 elements that gain H3K27ac later in differentiation drive gene activation at S4 (pancreatic progenitor stage) (cluster no. 6, Fig. 3c and Extended Data Fig. 9c). These results show that the unmasking of MER11 elements in the *ZNF808* KO impacts nearby gene expression in early differentiation with downstream effects as differentiation progresses.

We performed gene set enrichment analysis using Enrichr[27] and observed fetal and adult liver genes activated in *ZNF808* KO from S2 (primitive gut tube) onwards (fetal liver enriched at stage S2, false discovery rate (FDR) < $10^{-14}$, odds ratio (OR) 12.3) (Extended Data Fig. 9d). We tested whether this reflected the divergence between pancreatic buds and hepatic cords in vivo, with comparison to Carnegie stage 12–14 human embryos that correspond to our S2–S4 in vitro stages[28] (Fig. 4c, top). We found hepatic cord genes activated in our *ZNF808* KO

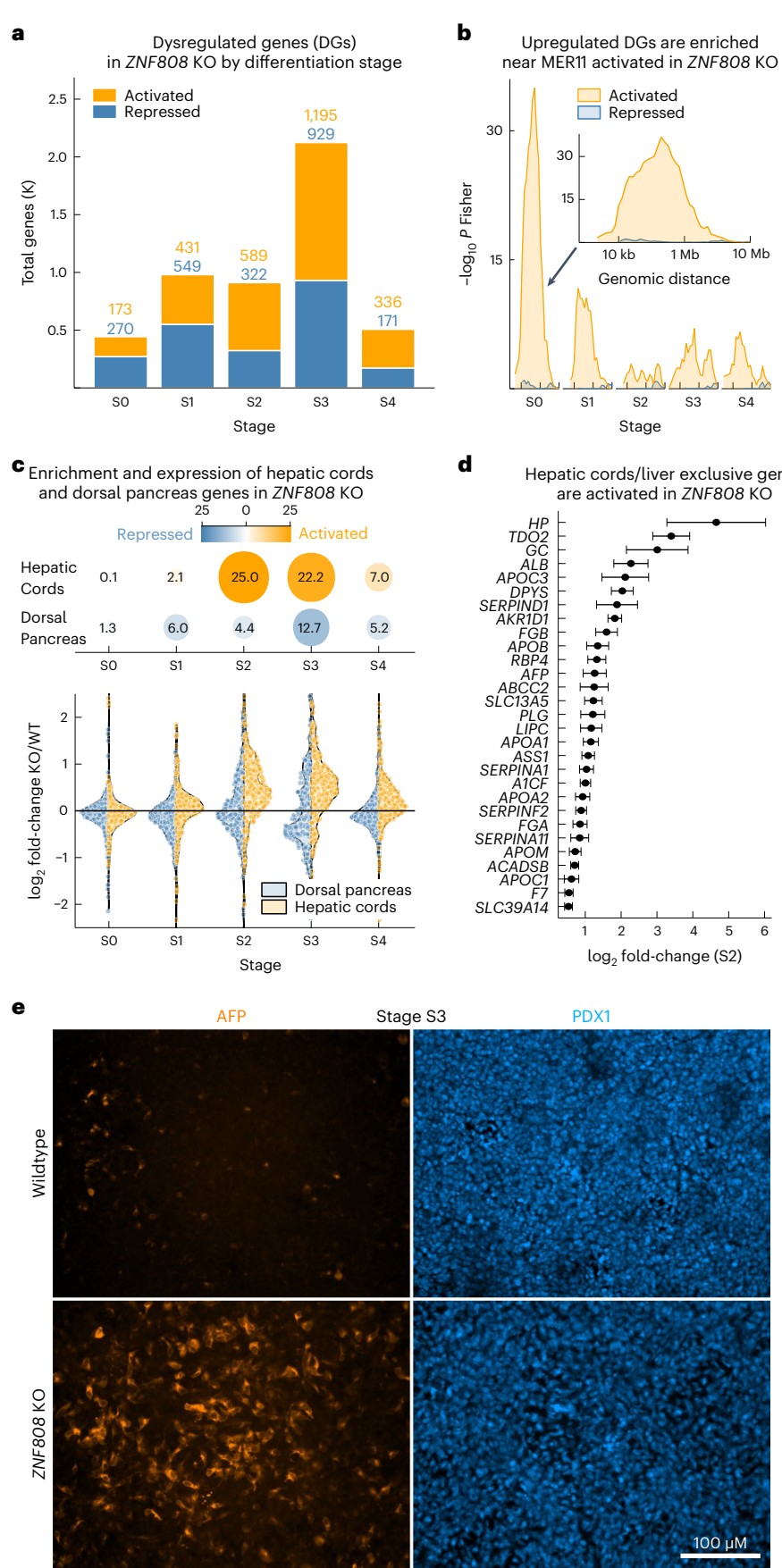

**a**, Dysregulated genes (DGs) in *ZNF808* KO by differentiation stage

**b**, Upregulated DGs are enriched near MER11 activated in *ZNF808* KO

**c**, Enrichment and expression of hepatic cords and dorsal pancreas genes in *ZNF808* KO

**d**, Hepatic cords/liver exclusive genes are activated in *ZNF808* KO

**e**, AFP    Stage S3    PDX1

**Fig. 4 | Loss of *ZNF808* during pancreas differentiation leads to activation of genes in proximity to unmasked MER11 elements and induction of a liver gene expression program.**
**a**, Loss of *ZNF808* leads to perturbed gene expression throughput pancreatic differentiation. Bar chart showing total genes activated and repressed with FDR < 0.05 and |FC| > 1.25 for each stage of pancreatic differentiation. K, thousands.
**b**, Dysregulated genes are found in proximity to unmasked MER11 elements in the *ZNF808* KO early in differentiation. Proximity enrichment ($-\log_{10}$ Fisher exact right-tail *P* value) showing an excess of dysregulated genes compared to all genes in proximity to MER11 elements losing H3K9me3 and gaining H3K27ac in the *ZNF808* KO as a function of distance between genes and binding sites for genes activated (orange) and repressed (blue) on $\log_{10}$ scale. Enrichment peaks between 10 kb and 100 kb suggest ZNF808 repressing distal gene enhancers. A total of 43.4% and 23.9% of activated genes at S0 and S1, respectively, are within 1 MB of a MER11 element.
**c**, Hepatic cords genes are activated and dorsal pancreas bud genes are repressed in *ZNF808* KO. Top, Fisher exact enrichment between *ZNF808* KO activated and repressed genes and genes more highly expressed in CS12–14 hepatic cords or dorsal pancreatic buds, respectively[28]. Bottom, $\log_2$ fold-change *ZNF808* KO over WT for dorsal pancreas bud (left, blue) and hepatic cords (right, orange) genes. Each dot represents a single gene. **d**, Genes exclusively expressed in liver and activated in hepatic cords are activated in *ZNF808* KO. The $\log_2$ fold-changes of the 29 genes that are expressed in CS12–14 hepatic cords, exclusively expressed in GTEx liver and activated at S2 in *ZNF808* KO. Error bars give DESeq2 standard error of the $\log_2$ fold-change, $n = 3$ independent biological replicates.
**e**, Immunostaining of AFP and PDX1 at S3 (posterior foregut) stage. Confirmation of RNA-seq results by immunostaining, showing activation of AFP in *ZNF808* KO in PDX1 positive cells (representative of three independent differentiation experiments; scale bar, 100 µm). Independent replicate stainings from the same cell lines are given in Extended Data Fig. 3d.

(FDR < $10^{-45}$, OR 6.4, stage S2) and conversely dorsal pancreas genes repressed (FDR < $10^{-12}$, OR 3.1, stage S3). The divergence between hepatic cord and dorsal pancreas genes peaks at posterior foregut stage S3 (Fig. 4c, bottom), coinciding with the introduction of factors modulating propancreatic signaling pathways (for example, retinoic acid and BMP and SHH inhibition); we suggest that this extinguishes the hepatic fate in the *ZNF808* KO.

Using GTEx we found that the induction of hepatic gene expression in the *ZNF808* KO was enriched in genes whose expression is exclusive to liver (Extended Data Fig. 10; $P < 10^{-16}$, OR 7.6, stage S2). We identified a set of 29 liver-exclusive, hepatic cord genes activated at S2, including genes such as *AFP*, *ALB*, *APOA1* and *LIPC* (Fig. 4d). This included an example of a gene (*TDO2*) activated in proximity to a MER11 element, suggesting that MER11 unmasking is compatible with hepatic gene expression (Extended Data Fig. 10c–e). We confirmed the transcriptomic upregulation of the early liver marker alpha-fetoprotein (AFP) by immunostaining (Fig. 4e and Extended Data Fig. 3d). Taken together, our data show that ZNF808 is essential to prevent a liver gene expression program from being aberrantly activated during pancreas differentiation, suggesting a potential mechanism for pancreatic agenesis.

We report the identification of recessive loss-of-function variants in *ZNF808*, a primate-specific gene, as a cause of pancreatic agenesis, a congenital developmental disorder. This confirms the crucial importance of gene discovery efforts in patients with extreme phenotypes.

Our study shows that ZNF808 is critical for human pancreatic development. Previous human genetic studies identified pancreatic developmental genes, including *GATA6* (refs. 1,29,30), *GATA4* (refs. 2,15,29–31) and *HNF4A* (refs. 16,32,33), with different dosage-dependent effects between mouse and humans, suggesting key differences in pancreatic development between the two species. Our identification of the role of ZNF808 and MER11 element regulation during human pancreatic development offers important insights into how mechanisms regulating pancreas development have diverged between primates and other mammals and supports a key role of ZNF808 in regulating differentiation of endoderm progenitors between liver and pancreas lineages. We provide correlative evidence that the unmasking of MER11 elements in the *ZNF808* KO ultimately leads to the downstream induction of a liver gene expression program during pancreas differentiation. Future work combining targeted genome editing and 3D conformation data will be necessary to confirm which of the regulatory regions repressed by ZNF808 are responsible for disrupting pancreas development in our patients.

Our characterization of the role of ZNF808 during early development offers insights into the role of KZFPs, showing that they provide a negative layer of regulation that masks transposable element regulatory potential in cellular settings where the right transcription factors are present, yet the domesticated regulatory activity is undesired. We believe that this is an evolutionary mechanism that allows for a broader range of transposons to be domesticated as they do not have to prove beneficial in all cellular contexts if KZFPs can selectively silence them where they would have a negative impact.

The crucial role of ZNF808 and MER11 elements in human pancreas development underscores that even primate-specific genes and transposable elements can be involved in important aspects of human biology. This discovery offers important insights into the evolution of gene regulation during human development and opens new avenues of research in the fields of human genetics and diabetes.

## Online content

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

## Pancreatic Agenesis Gene Discovery Consortium

**Wafaa Laimon[12], Samar S. Hassan[13], Mohamed A. Abdullah[14], Anders Fritzberg[15], Emma Wakeling[16], Nisha Nathwani[17], Nancy Elbarbary[18], Amani Osman[19], Hessa Alkandari[20,21], Abeer alTararwa[21], Abdelhadi Habeb[22], Abdulmoein Eid Al-Agha[23], Ihab Abdulhamed Ahmad[24], Majida Noori Nasaif Aldulaimi[25], Ala Ustyol[26], Hiba Mohammed Amin Binomar[27] & Mohammad Shagrani[27,28]**

[12]Pediatric Endocrinology and Diabetes Unit, Department of Pediatrics, Mansoura Faculty of Medicine, Mansoura University Children's Hospital, Mansoura University, Mansoura, Egypt. [13]Gaafar Ibn Auf Pediatric Tertiary Hospital, Khartoum, Sudan. [14]Faculty of Medicine, University of Khartoum, Khartoum, Sudan. [15]Region of Dalarna—Child and Adolescent Medicine, Falun Hospital, Falun, Sweden. [16]North East Thames Regional Genetic Service, Great Ormond Street Hospital for Children NHS Foundation Trust, London, UK. [17]Paediatric Department, Bedfordshire Hospitals NHS Foundation Trust, Luton and Dunstable Hospital Site, Luton, UK. [18]Diabetes Unit, Department of Pediatrics, Faculty of Medicine, Ain Shams University, Cairo, Egypt. [19]Imperial College London Diabetes Centre, Al Ain, United Arab Emirates. [20]Department of Population Health, Dasman Diabetes Institute, Kuwait City, Kuwait. [21]Department of Paediatrics, Farwaniya Hospital, Kuwait City, Kuwait. [22]Paediatric Department, Prince Mohammed bin Abdulaziz Hospital, Madinah, Saudi Arabia. [23]Paediatric Department, King Abdulaziz University Hospital, Jeddah, Saudi Arabia. [24]Pediatric Endocrinology Unit, Zagazig University Hospital, Zagazig, Egypt. [25]Fakeeh Care, Jeddah, Saudi Arabia. [26]Department of Pediatrics, University of Health Sciences Haseki Training and Research Hospital, İstanbul, Turkey. [27]Organ Transplant Centre, King Faisal Specialist Hospital and Research Center, Riyadh, Saudi Arabia. [28]College of Medicine, Alfaisal University, Riyadh, Saudi Arabia.

## Methods

### Subjects

The study was conducted in accordance with the Declaration of Helsinki and all subjects or their parents/guardian gave informed written consent for genetic testing. DNA testing and storage in the Beta Cell Research Bank was approved by the Wales Research Ethics Committee 5 Bangor (REC 17/WA/0327, IRAS project ID 231760). For Proband 2, parents gave written consent for collection of a skin biopsy to be dedifferentiated to iPSCs to study the patient's cause of neonatal diabetes. No participant received compensation for entering this study.

Individuals with neonatal diabetes diagnosed before the age of 6 months were recruited by their clinicians for molecular genetic analysis in the Exeter Genomics Laboratory. For one patient (case 10 in Supplementary Table 3), a *ZNF808* homozygous loss-of-function variant was identified by exome sequencing analysis by the Center for Genomic Medicine, King Faisal Specialist Hospital and Research Center, Riyadh.

Proband 1 and 2 were selected from a larger cohort of individuals with pancreatic agenesis[3]. Within this group, they were the only two individuals without a genetic diagnosis who were born to consanguineous parents.

### Genetic analysis

Exonic sequences were enriched from genomic DNA using Agilent's SureSelect Human All Exon kit (v.4) and then sequenced on an Illumina HiSeq 2000 sequencer using 100 base pair (bp) paired-end reads. The sequencing data were analyzed using an approach based on the GATK best-practice guidelines. GATK v.3.7 HaplotypeCaller was used to identify variants that were annotated using Alamut batch v.1.8 (human reference genome assembly hg19) and variants that failed the QD2 VCF filter or had less than five reads supporting the variant allele were excluded. Copy number variants were called by SavvyCNV[36], which uses read depth to judge copy number states. SavvyVcfHomozygosity[34] was used to identify large (>3 Mb) homozygous regions in the exome sequencing data (https://github.com/rdemolgen/SavvySuite).

A total of 232 patients diagnosed with diabetes before age 6 months in whom the known genetic causes of neonatal diabetes had been excluded were analyzed either by using a targeted next-generation sequencing assay, which includes baits for known neonatal diabetes genes and additional candidate genes followed up from gene discovery, such as *ZNF808*, or by independent genome sequencing analysis. Variant confirmation and cosegregation in family members were performed by Sanger sequencing (primers available on request).

### Cell culture and in vitro differentiation of human stem cells

The hESC (WA01/H1 line, Wicell) and patient-derived iPSC were cultured on Matrigel-coated plates (BD Biosciences) in Essential 8 (E8) medium (Life technologies, A1517001) and passaged using EDTA. To carry out the differentiation experiments, hPSC (human pluripotent stem cells) were dissociated using EDTA, then seeded on new Matrigel-coated plates in E8 medium supplemented with 10 µM rho-associated kinase inhibitor (ROCKi, catalog no. Y-27632; Selleckchem catalog no. S1049) at a density of 0.21 million cells per cm$^2$. After 24 h, the differentiation was started by washing the cells with PBS, then changing the medium to D0 medium. The differentiation was carried out using our optimized protocol as previously described[17].

### Genome editing

To create an in vitro model for studying the role of ZNF808 in pancreatic development, guide RNAs targeting the zinc fingers domain of the fifth exon of *ZNF808* for deletion were designed using Benchling (https://benchling.com) (gRNAs sequence available in Supplementary Table 7). The gRNAs with the highest quality score and lowest off-targets score were selected and purchased, alongside the RNP components (Alt-R Cas12a (Cpf1) Ultra protein, crRNA), from Integrated DNA Technologies (IDT) and used according to the manufacturer's recommended

protocol. A total of 2 million cells were electroporated with the RNP complex using Neon Transfection system (Thermo Fisher, 1100 V, 20 ms, two pulses) and plated on Matrigel-coated plates in E8 medium containing 10 µM ROCK inhibitor overnight. Afterwards, cells were single-cell sorted, expanded and screened for the desired deletion using PCR. Positive clones were validated by Sanger sequencing at Eurofins Genomics and the sequences were aligned using Geneious Prime 2020.1.1. The KO clones were characterized for pluripotency, chromosomal integrity and the top three off-target hits predicted by the online tool CRISPOR[37] were checked with no off-target indels found.

### Formaldehyde crosslinking

To fix the cells for ChIP–seq samples preparation, cells were incubated with TrypLE for 5–10 min at 37 °C and gently homogenized with the pipette, then pooled in a 15 ml Falcon tube containing warm DMEM. Afterwards, cells were spun down at 250$g$ at room temperature for 3 min, then resuspended in DMEM at a concentration of 5 million cells per ml and incubated with 333 mM fresh 16% methanol-free formaldehyde at room temperature for precisely 10 min. Formaldehyde was quenched using 250 mM Tris pH 8.0 for another 10 min at room temperature, then cells were spun down at 250$g$ at 4 °C for 5 min. Cell pellet was resuspended gently in PBS, aliquoted into 1.5 ml Eppendorf tubes and spun down at 250$g$ at 4 °C for 5 min. Supernatant was removed and samples were stored at −80 °C until further processing.

### Flow cytometry analysis

The hESC-derived cells were dissociated into single cells by incubation with TrypLE for 5–10 min at 37 °C and resuspended in cold FBS/PBS (5% v/v). For surface marker staining of CXCR4, 1 million cells were incubated with the directly conjugated antibody CD184/CXCR4 APC at a final dilution of 1:10 for 30 min at room temperature. For intracellular markers, 1 million cells were first fixed in 350 µl of Cytofix/Cytoprem Buffer (BD, no. 554722) for 20 min at 4 °C, then washed twice with BD Perm/Wash Buffer Solution (BD, no. 554723). Cell pellet was resuspended in 80 µl of FBS/BD Prem/Wash buffer (4% v/v) and incubated with the corresponding directly conjugated antibody at a final dilution of 1:80 overnight at 4 °C. After incubation with the antibody, cells were washed twice and analyzed using FACSCalibur cytometer (BD Bioscience), BD CellQuest Pro software v.4.0.2 and FlowJo software v.9 (Tree Star). Details of the antibodies are listed in Supplementary Table 6.

### Immunocytochemistry

For adherent cultures, cells were fixed in 4% PFA for 15 min at room temperature, permeabilized with 0.5% Triton X100 in PBS, then blocked with UltraV block (Thermo Fisher) for 10 min and incubated with primary antibodies diluted in 0.1% Tween in PBS overnight at 4 °C. After incubation, cells were washed twice with PBS and incubated with corresponding secondary antibodies diluted in 0.1% Tween in PBS for 1 h at room temperature. Details of the antibodies are listed in Supplementary Table 6.

### Chromatin immunoprecipitation

The following steps were performed with ice-cold samples and buffers containing a protease inhibitor cocktail (cOmplete ULTRA Tablets EDTA-free, Roche). In 1.5 ml DNA LoBind tubes, 4 million fixed cells were resuspended in 1 ml of lysis buffer 1 (50 mM HEPES-KOH pH 7.4, 140 mM NaCl, 1 mM EDTA, 0.5 mM EGTA, 10% glycerol, 0.5% NP40, 0.25% Triton X100, proteinase inhibitor 1×) incubated at 4 °C on a rotating wheel at 10 r.p.m. for 10 min. Cells were then centrifuged at 1,700$g$ for 5 min at 4 °C. Supernatant was discarded and pellets were resuspended in 1 ml of lysis buffer 2 (10 mM Tris HCl pH 8.0, 200 mM NaCl, 1 mM EDTA, 0.5 mM EGTA, proteinase inhibitor 1×) and incubated at 4 °C on a rotating wheel at 10 r.p.m. for 10 min. After centrifugation at 1,700$g$ for 5 min at 4 °C, supernatant was discarded and pellets were washed with 500 µl of SDS shearing buffer (1 mM Tris HCl pH 8.0, 1 mM EDTA,

0.15% SDS, proteinase inhibitor 1×), without disturbing the pellets, followed by centrifugation at 1,700g for 5 min. Washing was repeated twice and pellets were resuspended in 1 ml of SDS shearing buffer and transferred into Covaris milliTUBE 1 ml AFA Fiber. Chromatin was sheared on a Covaris E220 for 6 min at 5% duty cycle, 140 W, 200 cycles. The sheared chromatin was then transferred into 1.5 ml DNA LoBind tubes and centrifuged at 10,000g for 5 min at 4 °C. Supernatant was then used immediately for immunoprecipitation. Chromatin quality control was performed on Bioanalyzer 2100 (Agilent) to verify that most fragments ranged between 200 and 600 bp.

For H3K9me3 IP, chromatin corresponding to 1 million cells was put in a new 1.5 ml DNA LoBind tube and topped to 900 μl with SDS shearing buffer. For H3K27ac IP, chromatin corresponding to 3 million cells was put in a new 1.5 ml DNA LoBind tube and topped to 900 μl total with SDS dilution buffer (1 mM Tris HCl pH 8.0, 1 mM EDTA, 0.026% SDS, proteinase inhibitor 1×) so that the SDS in the final IP buffer would be 0.1%. IP conditions were further adjusted to 150 mM NaCl and 1% Triton final, 1 ml final volume. Either 1 μl of H3K9me3 antibody (catalog no. 39685, Active Motif) or 5 μg of H3K27ac antibody (catalog no. 39161, Active Motif) was added. The IP was then incubated on rotating wheel at 10 r.p.m. at 4 °C overnight. The next day, Dynabeads Protein G (5 μl for the H3K9me3 IP or 25 μl for the H3K27ac) were put on magnet and supernatant was removed. Beads were then resuspended in the full volume of the IP and incubated for 2 h on a rotating wheel at 10 r.p.m. at 4 °C.

For H3K9me3, low-salt washing buffer (10 mM Tris HCl pH 8.0, 150 mM NaCl, 1 mM EDTA, 1% Triton X100, 0.15% SDS, 1 mM PMSF) and high-salt washing buffer (10 mM Tris HCl pH 8.0, 500 mM NaCl, 1 mM EDTA, 1% Triton X100, 0.15% SDS, 1 mM PMSF) were used.

For H3K27ac, low-salt washing buffer (20 mM Tris HCl pH 8, 150 mM NaCl, 2 mM EDTA, 1% Triton X100, 0.1% SDS, 1 mM PMSF) and high-salt washing buffer (20 mM Tris HCl pH 8, 500 mM NaCl, 2 mM EDTA, 1% Triton X100, 0.1% SDS, 1 mM PMSF) were used.

All washes took place while IPs and buffers were ice cold. PMSF was always added in the buffers immediately before each wash. The IPs were placed on a magnetic rack and supernatant was discarded. Beads were resuspended in low-salt washing buffer and transferred into a clean DNA LoBind tube. Beads were then placed on a magnetic rack; supernatant was removed and beads were resuspended in low-salt washing buffer. The mixture was placed again on a magnetic rack, supernatant was discarded and beads were washed with high-salt washing buffer. Once more, samples were placed on the magnetic rack, supernatant was removed and beads were resuspended in LiCl buffer (10 mM Tris HCl pH 8.0, 1 mM EDTA, 0.5 mM EGTA, 250 mM LiCl, 1% NP40, 1% NaDOC, 1 mM PMSF). The mixture was placed on the magnetic rack, supernatant was removed and beads were washed with 10 mM Tris HCl pH 8.0 and transferred to a clean DNA LoBind tube. Finally, with the samples on the magnetic rack the supernatant was completely removed and beads were resuspended in elution buffer (10 mM Tris HCl pH 8.0, 1 mM EDTA, 1% SDS and 150 mM NaCl).

RNase A was added to the elution buffer at a final concentration of 0.5 μg μl⁻¹ and samples were incubated at 37 °C for 1 h in a shaking incubator at 1,100 r.p.m. Subsequently proteinase K was added at a concentration of 400 ng μl⁻¹ and chromatin was decrosslinked at 65 °C overnight. The supernatant was collected and purified using Serapure beads before library preparation.

To control for the efficiency of the IP, we used qPCR with primers targeting negative and positive regions in the genome for the histone marks H3K9me3 and H3K27ac, respectively (primer sequences available in Supplementary Table 7).

**Library preparation.** Libraries were prepared using NEBNext Ultra II DNA Library Prep Kit for Illumina following the manufacturer's instructions. Adapters and indexed primers design, resuspension and annealing were as previously described[38]. The adapters used were iTrusR2-stubRCp and iTrusR1-stub.

The library was quantified by qPCR using KAPA SYBR FAST and the set of primers itru7_101_01 and itru5_01_A. After quantification the library was amplified and double indexed (primers details in Supplementary Table 7). Amplified libraries were double size selected using home-made Serapure beads to enrich for fragments between 200 and 600 bp.

After amplification, confirmation that the IP remained efficient was carried out using qPCR with primers targeting genomic regions negative and positive regions for the histone marks H3K9me3 and H3K27ac, respectively, as mentioned above. Libraries were sent for 150 bp paired-end sequencing at Novogene.

**ChIP–seq analysis.** Reads from each library were mapped on hg19 using Bowtie2 v.2.4.4 (ref. 39) with the 'very-sensitive-local' setting. Mapped reads were compressed in BAM files and indexed using SAMtools v.1.18 (ref. 40). We used MACS2 v.2.2.7.1 (ref. 41) to call peaks using the corresponding input datasets as background controls. Peaks were excluded if the average read had MAPQ < 20. Reads intersecting intervals were counted with HTSeq 0.13.5, with a filter of MAPQ > 20 for properly paired reads applied. To identify MER11 elements losing H3K9me3 and gaining H3K27ac, we took peak regions defined by an overlap with H3K9me3 (pybedtools 0.8.1 and bedtools v.2.30.0, $f = 0.5$, $F = 0.5$; e, True; u, True; wa, True) in WT cells with no overlap in the KO and the opposite for H3K27ac to derive a final list of 220 MER11 elements activated in *ZNF808* KO. To identify epigenetic responses over differentiation, we quantified depth normalized reads in each region and standardized H3K9me3 and H3K27ac signals separately by applying a $z$-score normalization (zero-meaned and standard deviation scaled) to each region over the differentiation stages. To assign regions to clusters of similar behavior, we performed $k$-means clustering with parameters n_clusters = 6, init = 'random', max_iter = 3,000, n_init = 100 using Scipy. To visualize these, we also performed hierarchical clustering using fastcluster v.1.2.4 with optimal leaf ordering which we mapped onto our $k$-means clustering to provide a heatmap with consistent within cluster and between cluster ordering. ChIP–seq tables and data analyzed in Python used pandas 1.3.5 and NumPy 1.12.5. Enrichment of transcription factors was by downloading hg19 ChIP peak data from the repository ChIP-Atlas (https://chip-atlas.org[14]). We used bedtools fisher v.2.30.0 with options −f 0.5 -F 0.5 -e to calculate intersection contingency table between MER11 elements of distinct subfamilies (MER11A, MER11B and MER11B) and each ChIP-Atlas dataset. We then calculated right-tail Fisher exact $P$ values for over-representation and ORs using the Julia Distributions.jl package and applied Benjamini−Hochberg FDR control using MultipleTesting.jl.

Analysis of chromatin accessibility at MER11 elements was performed using the chromatin atlas[25]. Peak calls of chromatin accessibility for each cell type deconvoluted from single-cell signal clustering were downloaded and overlapped with bedtools with $f = 0.5$, $F = 0.5$; e, True (at least 50% overlap from either peak).

Analysis of the epigenetic status of MER11 in various cell types was done using the expanded NIH Roadmap analysis by ref. 26. Only datasets with measured H3K9me3 and at least one of H3K27ac or H3K4me1 were retained. Chromatin states were assigned by overlapping with precalculated HMM states from ref. 26 and were collapsed by cell type using the provided metadata—overlap in at least one cell type with a particular chromatin state was counted as present and Enhancer/TSS status were given precedence over heterochromatin if both were found in a cell type at a particular locus in two different biosamples.

**Gene expression analysis by RNA-seq**
Stranded, poly-A selected RNA-seq libraries were prepared and sequenced from three independent replicates of our differentiation time course (paired-end 150 bp reads) by Novogene.

Stranded paired-end 150 bp RNA-seq reads were aligned to the hg19 genome using STAR v.2.7.3a (ref. 42) and quantified against Gencode v.36 release liftover to hg19 by RSEM v.1.3.2 (ref. 43) using

the RSEM-STAR pipeline, with further options --seed 1618 --calc-pme --calc-ci --estimate-rspd --paired-end. RSEM estimated read counts per sample were rounded for use with DESeq2 v.1.30.1 (ref. [44]). We perform differential expression analysis for each stage WT versus *ZNF808* KO for all genes with at least ten raw counts in all replicates of one condition. Gene expression varies substantially over the differentiation time course with subsets of genes only expressed at early or late stages, therefore when testing each stage, we supply DESeq2 with samples from adjacent stages for information sharing in estimating dispersions. We have three independent replicates of our differentiation time course and we perform a paired analysis between WT and KO pairs within DESeq2 with the model ~ExpNum + Genotype, where ExpNum is a factor indicating the replicate and GenoType is a factor indicating WT or *ZNF808* KO, we then perform a contrast between the two levels of the GenoType factor and calculate differential expression with independentFiltering=FALSE. We consider genes with FC (fold change) > 1.25 and FDR < 0.05 differentially expressed.

To determine enrichments of differentially expressed genes in proximity to clusters of epigenetic response in *ZNF808* loss, we used the package ProximityEnrichment.jl (https://github.com/owensnick/ProximityEnrichment.jl). Specifically, we calculate hypergeometric right-tail $P$ values for the association of differentially expressed genes within $x$ bp of ZNF808-bound region cluster against a background of genes for $x$ in $[1, 1 \times 10^6]$. For proximity-enrichment heatmaps, we take the maximal enrichment from this interval.

To determine gene set enrichments for sets of differentially expressed genes, we use the Enrichr API[27], with the package (https://github.com/owensnick/Enrichr.jl) to recover enrichments for BioPlanet 2019, GO_Biological_Process 2018, GO_Cellular_Component 2018, GO_Molecular_Function 2018, Human_Gene_Atlas and KEGG_2019_Human gene sets. As Enrichr calculates enrichments using a generic background, we downloaded all definitions of Terms and Gene sets and recalculated Fisher exact right-tail $P$ values and ORs (using HypothesisTesting.jl) for terms over-represented in the dysregulated genes against a background of all genes tested for differential expression, we then corrected these for multiple testing using the Benjamini–Hochberg method (using MultipleTesting.jl). We report all enrichments in Supplementary Table 5, we selected the most prominent and relevant. To assess the intersection between our data and the laser capture of human embryo hepatic cords and dorsal pancreas[28], we took the set of genes detected in each stage of our data and hepatic cords versus dorsal pancreas and calculated the association between direction of dysregulation in our data with the direction of differentially expressed genes in the hepatic cords versus dorsal pancreas comparison with FDR < 0.05.

To identity genes exclusively expressed in the adult liver, we downloaded v8 gene level transcripts per million over all GTEx samples from https://gtexportal.org (file GTEx_Analysis_2017-06-05_v8_RNASeQCv1.1.9_gene_tpm.gct.gz). We calculated the IQR for all genes in all tissues and took a pragmatic definition of liver-exclusive expression: we took those genes for which the lower quartile of liver expression exceeded the upper quartile of expression in all other tissues, yielding 357 genes (Extended Data Fig. 10a). The genotype-tissue expression (GTEx) project was supported by the of the Office of the Director of the National Institutes of Health (NIH) and by NCI, NHGRI, NHLBI, NIDA, NIMH and NINDS. The data used for the analyses described in this manuscript were obtained from the GTEx Portal on 6 January 2023.

Data and code to perform transcriptomic analysis and to generate figure panels are available from https://github.com/owensnick/ZNF808Genomics.jl.

## Statistics and reproducibility

No statistical method was used to predetermine sample size for the patient cohort as the aim of the study was to identify the genetic cause of pancreatic agenesis, which is a rare monogenic condition.

Three independent replicates of H1 embryonic stem cell-derived WT and *ZNF808* KO differentiation time courses were collected. For transcriptomic analysis we used $n = 3$ replicates for WT and *ZNF808* KO and $n = 1$ for the patient-derived iPSC. From each of WT, *ZNF808* KO and patient-derived iPSC, one replicate per stage was assayed for H3K9me3 and H3K27ac epigenomics.

No data were excluded from any of the experiments described.

## Reporting summary

Further information on research design is available in the Nature Portfolio Reporting Summary linked to this article.

## Data availability

Clinical and genotype data are available only through collaboration as this can be used to identify individuals and so cannot be made openly available. Requests for collaboration will be considered following an application to the Genetic Beta Cell Research Bank (https://www.diabetesgenes.org/current-research/genetic-beta-cell-research-bank/). Contact by email should be directed to A. Hattersley (A.T.Hattersley@exeter.ac.uk). All requests for access to data will be responded to within 28 days. Transcriptomic and epigenomic data for WT and *ZNF808* KO are available from the NCBI Gene Expression Omnibus under accession GSE205164. Source data are provided with this paper.

## Code availability

Code and software versions used to analyze the data presented are indicated in the Methods and provided in https://github.com/owensnick/ZNF808Genomics.jl with persistent Zenodo https://doi.org/10.5281/zenodo.8375708.

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

## Acknowledgements

We are grateful to the patients and their families for taking part in our gene discovery study. We also thank S. Eurola, J. Ustinov and R. Ward for expert technical assistance. This work was supported by a Wellcome Trust Collaborative Award in Science to E.D.F., N.D.L.O., T.O., A.T.H. and M.I. (grant no. 224600/Z/21/Z). E.D.F. is a Diabetes UK RD Lawrence Fellow (19/005971) and has been the recipient of an EFSD Rising Star fellowship during this study. A.T.H. and S.E. were the recipients of a Wellcome Trust Senior Investigator award (grant no. WT098395/Z/12/Z) during this study and A.T.H. is employed as a core

member of staff in the NIH Research-funded Exeter Clinical Research Facility and is an NIHR Emeritus Senior Investigator. M.I. has a Sir Henry Dale Fellowship jointly funded by the Wellcome Trust and the Royal Society (grant no. 206688/Z/17/Z). S.E.F. has a Wellcome Trust Senior Research Fellowship (grant no. 105636/Z/14/Z). N.D.L.O. has a lectureship funded by a Research England's Expanding Excellence in England (E3) award. M.B.J. and M.N.W. are recipients of an Exeter Diabetes Centre of Excellence Independent Fellowship funded by E3. The work was supported by the National Institute for Health Research (NIHR) Exeter Biomedical Research Centre, Exeter, UK. R.E.J. is a Diabetes UK Harry Keen Clinician Scientist fellow (20/0006263) and received an Academy of Sciences Starter grant during this project. N.A.H. is funded by the Medical Research Council (grant nos. MR/000638/1 and MR/S036121/1). Most of the experimental studies were funded by the Academy of Finland Center of Excellence MetaStem (grant no. 312437), the Novo Nordisk Foundation (grant no. 0057286) and the Sigrid Juselius Foundation. H.M. is a member of the Doctoral Program in Integrative Life Science at University of Helsinki. A.T.'s doctoral studies were funded by the Foundation for Education and European Culture (IPEP) in Greece and by the Cambridge Trust. D.B. received funding from a European Molecular Biology Organization long-term fellowship (ALTF 295-2019). King Salman Center for Disability Research funded exome sequencing analysis for one case through Research Group no. RG-2022-010. For the purpose of open access, the author has applied a CC BY public copyright licence to any author accepted manuscript version arising from this submission.

## Author contributions

E.D.F., N.D.L.O., T.O., A.T.H. and M.I. designed the study. E.D.F., M.N.W., S.E.F., R.C., S.E., S.M. and M.J.B. performed the genetic analysis. M.N.W., C.F.W., N.D.L.O. and M.I. performed the conservation analysis. E.D.F., A.T.H., F.S.A. and the Pancreatic Agenesis Gene Discovery Consortium analyzed the clinical data. R.E.J. and N.A.H. provided human embryo RNA-seq data. N.D.L.O. and M.I. analyzed the transcriptomic and epigenetic data, with J.A.K. contributing to transcriptome analysis. H.M., J.S.-V., H.I., D.B. and S.M. performed the genome editing and differentiation studies in hESCs. A.T. performed the ChIP–seq assay. E.D.F., N.D.L.O., H.M., T.O., A.T.H. and M.I. wrote the first draft of the manuscript. All authors edited the paper.

## Competing interests

The authors declare no competing interests.

## Additional information

**Extended data** is available for this paper at https://doi.org/10.1038/s41588-023-01565-x.

**Correspondence and requests for materials** should be addressed to Timo Otonkoski, Andrew T. Hattersley or Michael Imbeault.

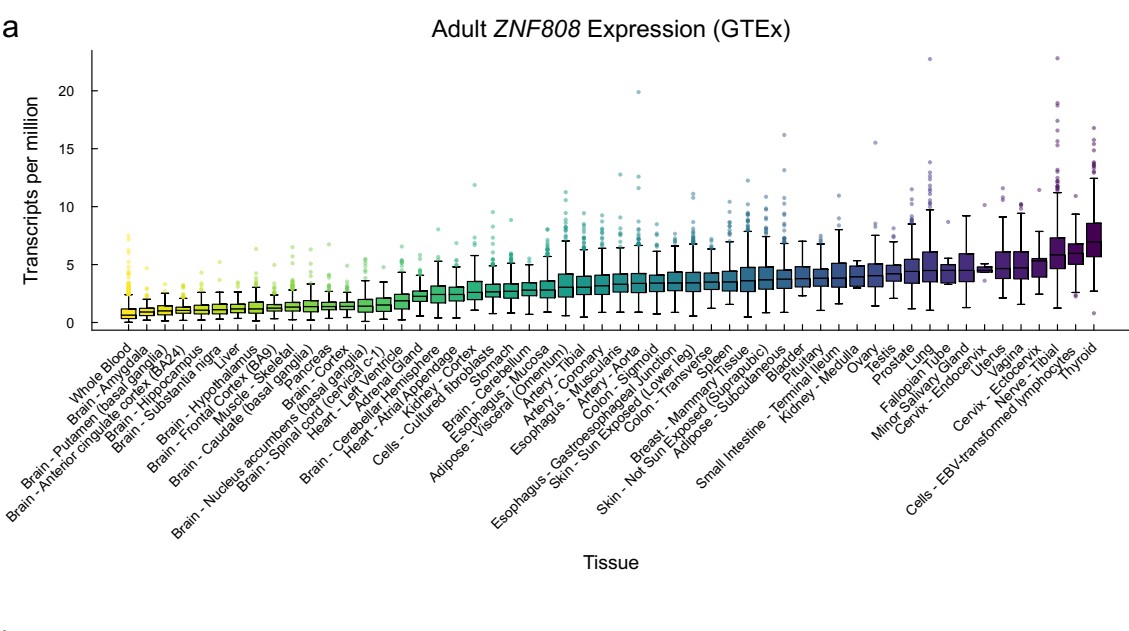

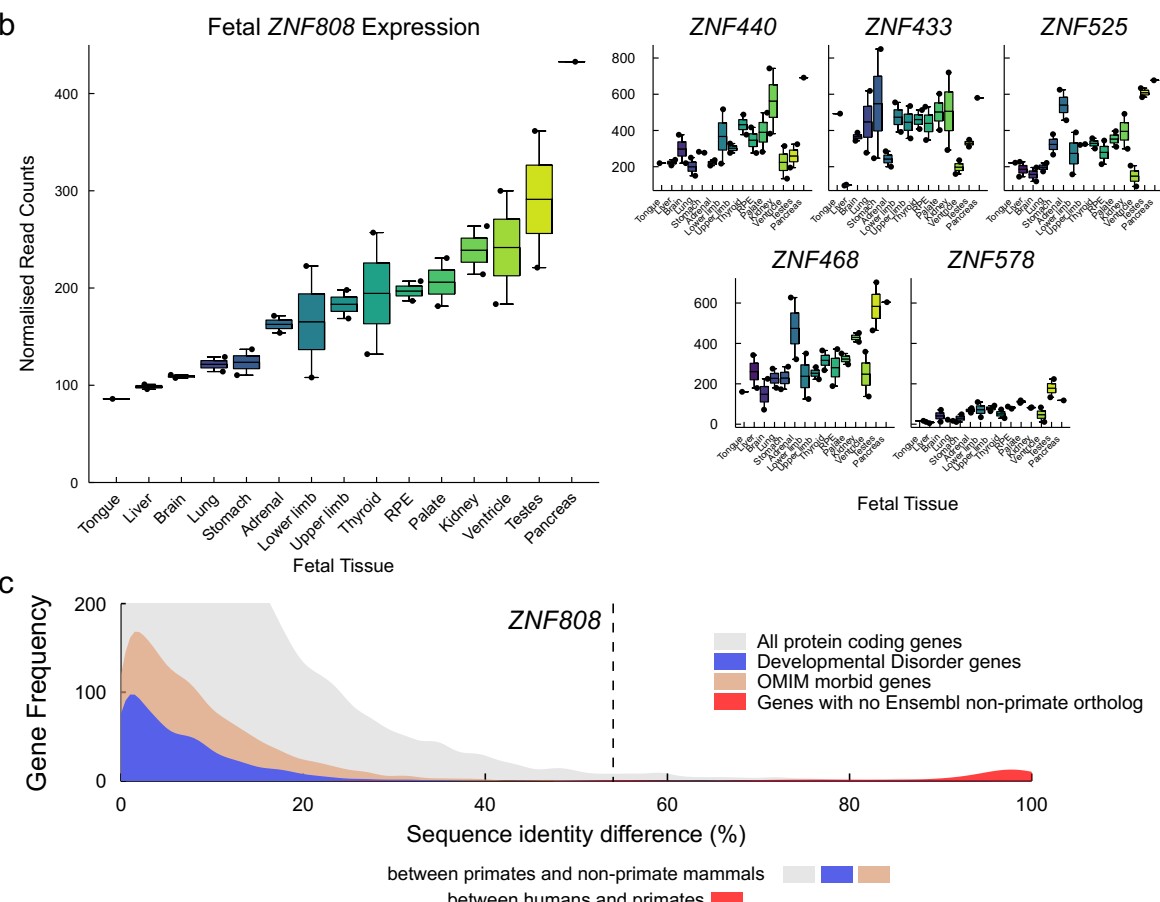

**Extended Data Fig. 1 | See next page for caption.**

**Extended Data Fig. 1 | The ubiquitously expressed *ZNF808* is the first primate-specific gene confirmed to cause a congenital developmental disease.**
**a**. *ZNF808* is expressed across GTEx adult tissues. *ZNF808* is expressed at variable levels across adult tissues with no tissue absent in expression. Boxplots describe *ZNF808* expression in the GTEx (Genotype-Tissue Expression) project. Data shows sum of GTEx v8 isoform level transcripts per million data calculated with RSEM. Boxplot central line denotes median, box limit the interquartile range and whiskers extend to furthest point within 1.5 interquartile range, data points are outliers exceeding whiskers, median sample size n = 291. **b**. *ZNF808* is maximally expressed in the embryonic pancreas and minimally expressed in the embryonic liver. Human embryo RNA-seq spanning CS14-22[5] (data normalized as in original publication) for all MER11-binding KZFPs, tissues ordered by expression and mean of replicates shown. Dots give expression for all replicates, central line of boxplots denotes median, box marks interquartile range and whiskers 1.5x interquartile range. **c**. *ZNF808* is the only primate-specific gene confirmed to cause a congenital developmental disease. Homology scores (including 1-to-1 and 1-to-many orthologues) for every protein-coding gene (Ensembl Biomart, 10th February 2020, release 98) for 26 primates and 70 non-primate mammals. The difference between the maximum % identity difference across all primates versus the maximum across all non-primates to human was calculated for each gene. Frequency densities (density estimations scaled to group size at 0.5% bin size) shown. Genes without a non-primate ortholog are plotted as percent identity between humans and primates (red), equivalent to a gene with 0% identity between primates and non-primates. Genes are grouped into non-disease causing (not present in OMIM-morbid; gray), disease causing (present in OMIM-morbid; green) and genetic causes of developmental disorders (present on DDG2P; blue). *ZNF808* is highlighted (dotted line, *ZNF808* is erroneously annotated as having a non-primate ortholog and its homology between primates and non-primates given). All OMIM-morbid and DDG2P genes with sequence identity difference >40 were manually checked with no evidence of a primate-specific disease gene causing a congenital developmental disorder found (Supplementary Table 4).

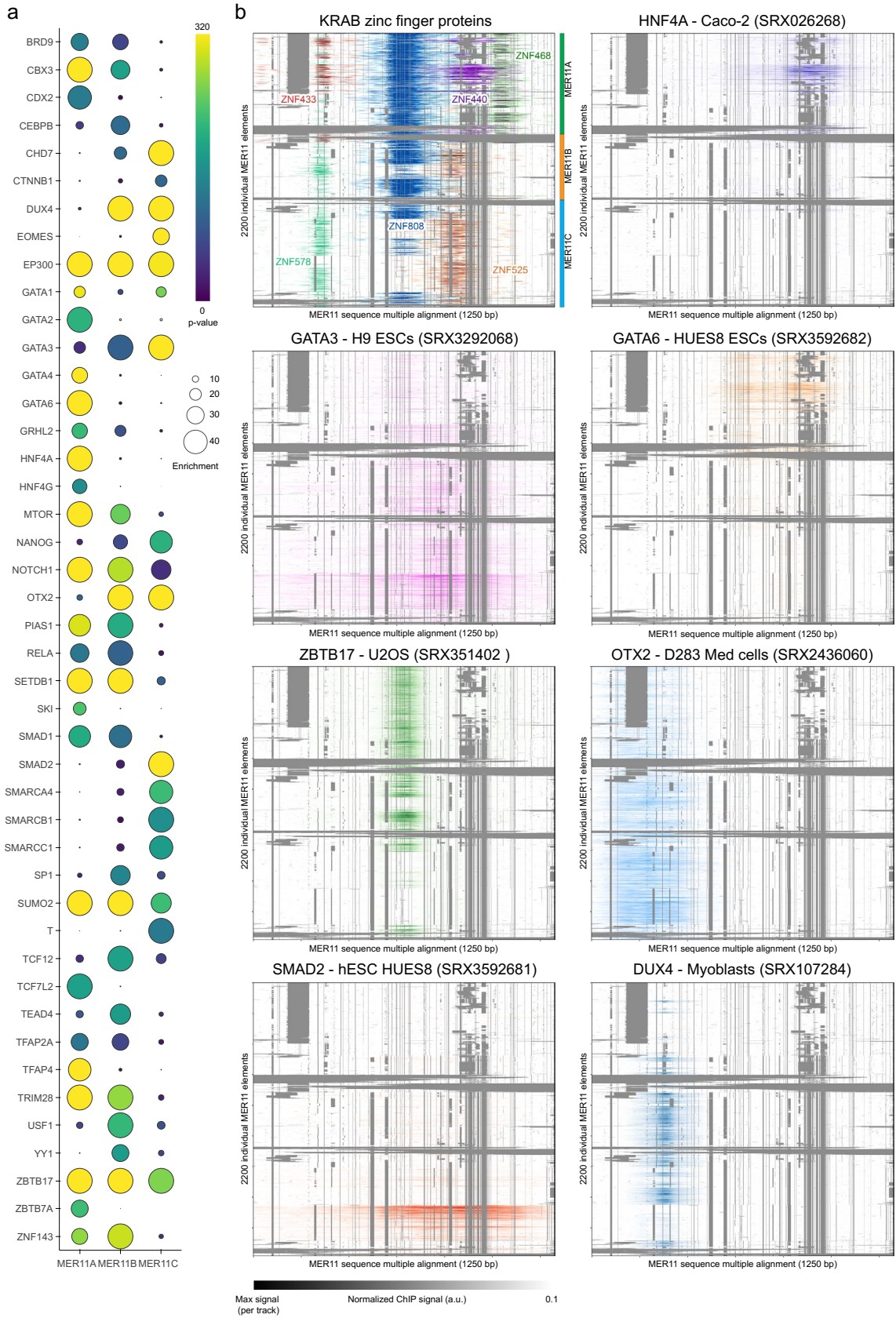

**Extended Data Fig. 2 | See next page for caption.**

**Extended Data Fig. 2 | MER11 elements subsets are enriched for various transcription factors. a.** Full list of hits from the ChIP-Atlas database with Fisher exact test right-tail p-value < 1e-100 enriched in various MER11 subfamilies, without multiple comparison adjustment. The color of each bubble is scaled with p-value and the radius with enrichment. If multiple experiments were found to be enriched for any given factor only the most significant value is shown.
**b.** Signal from selected factors overlay on a multiple alignment of MER11 sequences to show various subsets of sequences being targeted by specific transcription factors and KZFPs. Top left - Overlay of signal from published KZFPs ChIP–seq[8] on a multiple alignment of MER11A, MER11B and MER11C elements reveal that ZNF808 binds strongly in the centre of these elements. Five other KZFPs can be found on smaller subsets of elements - a clear pattern of semi-exclusive binding between ZNF808 and ZNF525 / ZNF578 is visible. All KZFPs binding MER11 elements represented here were found to be primate-specific at various levels. Multiple alignments generated with MAFFT v7.475 as performed in[8].

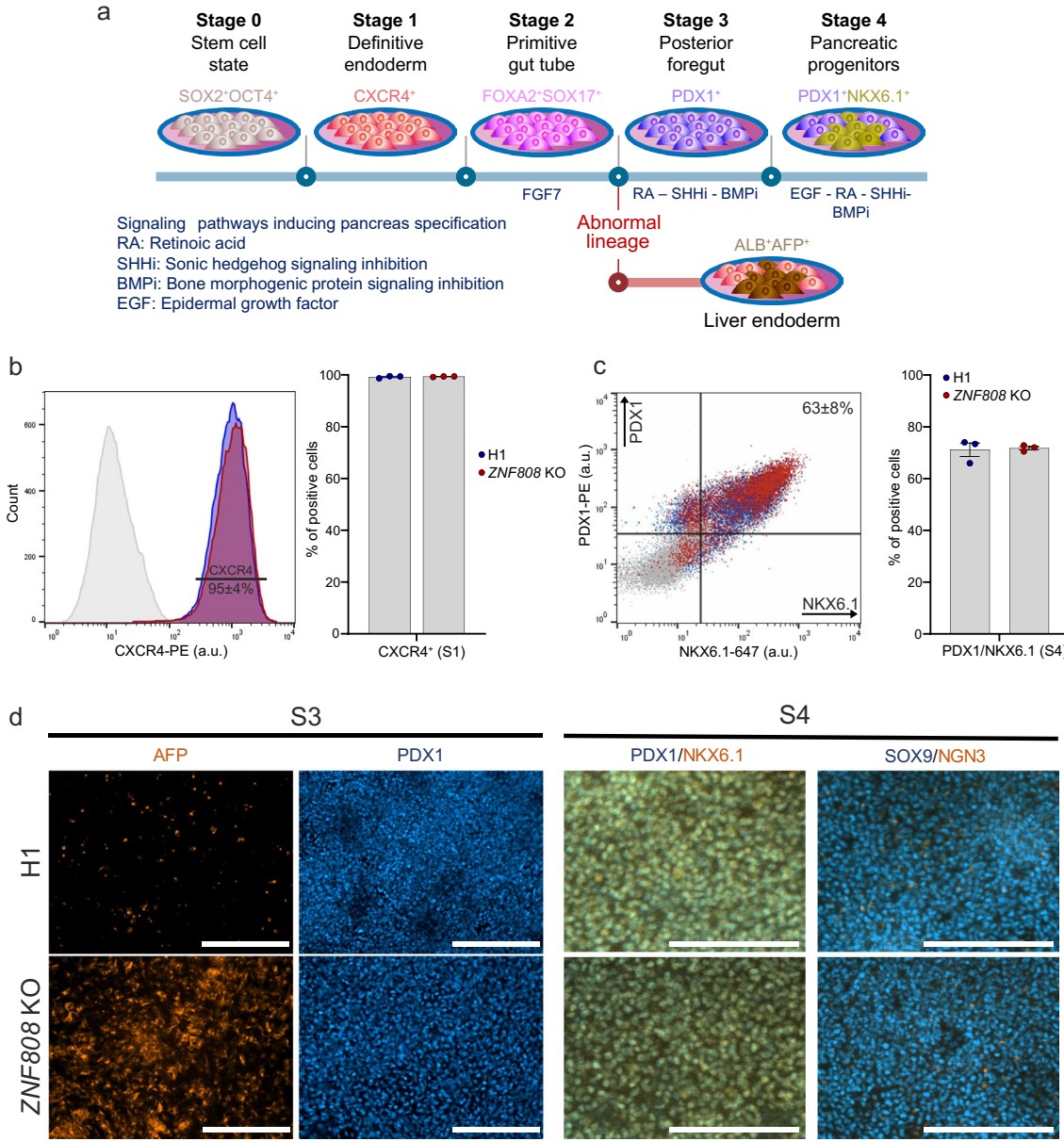

**Extended Data Fig. 3 | *ZNF808* KO and protocol of differentiation toward beta cells. a.** Differentiation protocol to generate pancreatic endoderm, progenitors and islets from human embryonic stem cells. **b.** Flow cytometry analysis for the definitive endoderm marker CXCR4. n = 3 independent differentiation experiments, data are presented as mean values ± SEM. Axis labels state the marker and fluorochrome used. (a.u.=arbitrary unit, S1=definitive endoderm stage). **c.** Flow cytometry analysis for the pancreatic progenitors markers PDX1 and NKX6-1 at S4 (pancreatic progenitors stage). n = 3 independent differentiation experiments, data are presented as mean values ± SEM. Axis labels state the marker and fluorochrome used. (a.u.=arbitrary unit). **d.** Immunohistochemistry analysis of S3 (posterior foregut stage) monolayer cells for PDX1 and alpha-fetoprotein (AFP); and S4 (pancreatic progenitors stage) monolayer cells for PDX1, NKX6-1, SOX9 and NGN3 (representative of 3 independent differentiation experiments, scale bar = 200 um).

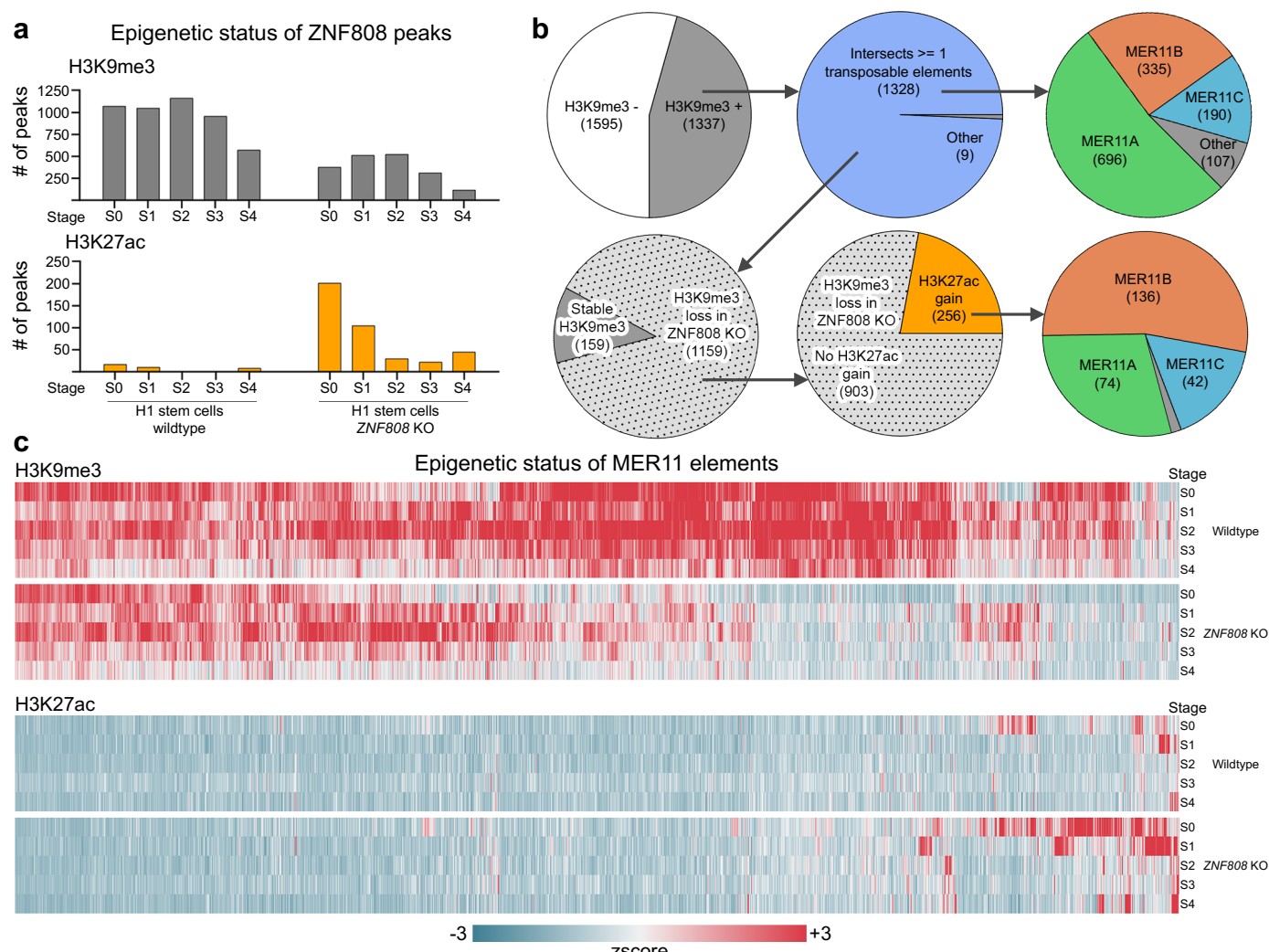

**Extended Data Fig. 4 | Epigenetic profile of ZNF808 peaks. a.** Analysis of H3K9me3 ChIP–seq peaks intersecting with ZNF808 peaks reveals that the majority of sites are covered in heterochromatin-associated H3K9me3 at early differentiation stages in the wild type. There is a partial loss of H3K9me3-positive loci in the *ZNF808* KO in stem cells and during differentiation. Analysis of H3K27ac reveals that a few sites are positive in wild-type cells but many more gain activity in the *ZNF808* KO at all stages, particularly at S0 and S1. **b.** Analysis of ZNF808 peaks shows that around half have H3K9me3 status in at least one stage of differentiation and that the vast majority of those are transposable elements of the MER11 family. Amongst those, the majority loses H3K9me3 in at least one differentiation stage

in the *ZNF808* KO – 21.7% of those also gain H3K27ac. These are similarly highly enriched in MER11 elements, although with a different distribution of subfamilies. Interval intersections were performed using pybedtools 0.81 using repeats annotation from repeatmasker.org for hg19, version RepeatMasker open-4.0.5 - Repeat Library 2014013I. Promoter coordinates were downloaded from Ensembl Biomart and extended 2.5 kb from the start site of all protein-coding genes. **c.** Heatmap showing H3K9me3 and H3K27ac signals over all MER11 elements in wild type and *ZNF808* KO during differentiation. Different patterns of loss of H3K9me3 and gain of H3K27ac are visible. Color scale is based on z-score per element of reads with MAPQ > 20, ranging from +2 (red) to −2 (blue).

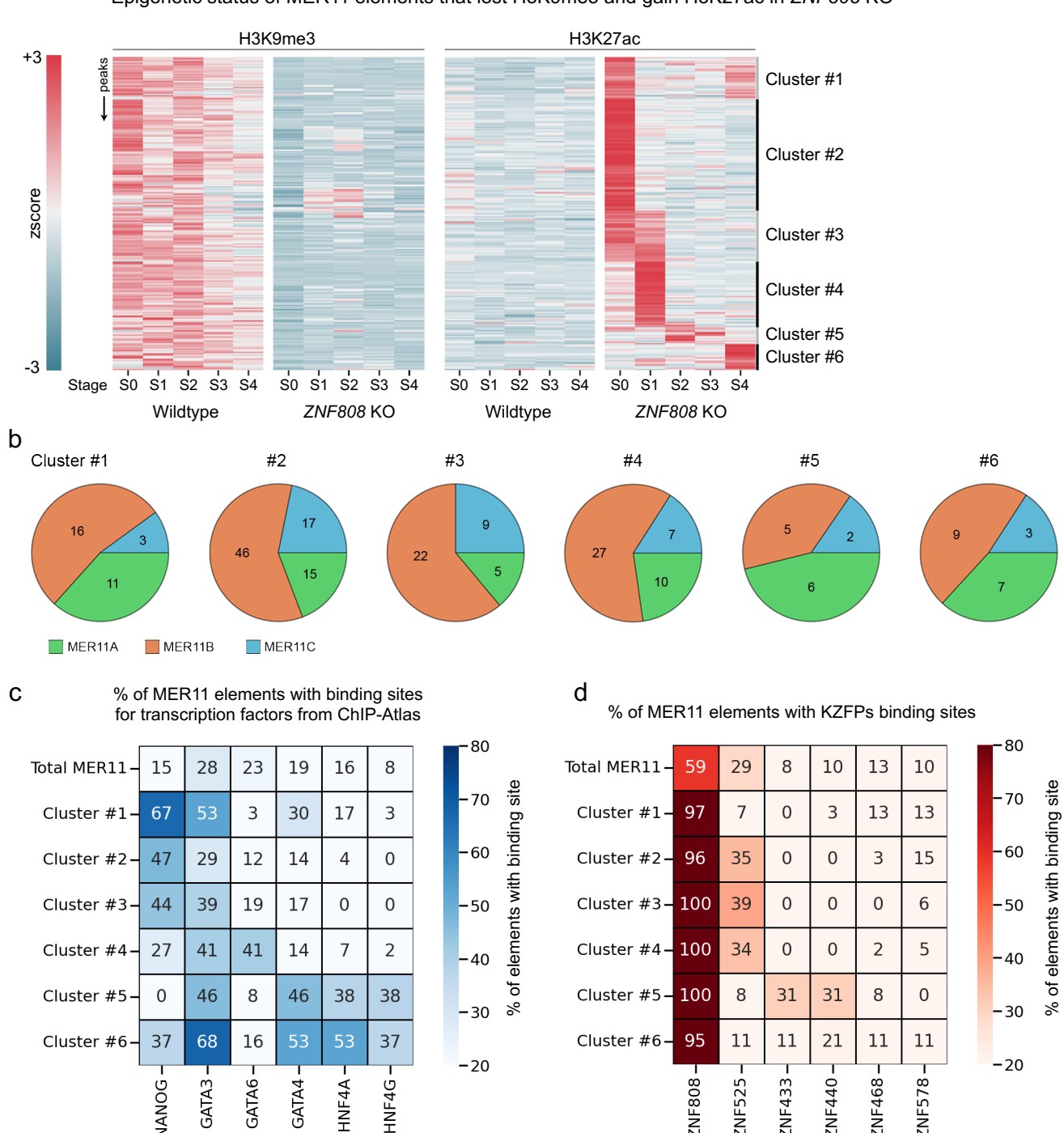

**Extended Data Fig. 5 | Dynamic epigenetic clusters of MER11 elements are enriched in transcription factors matching the stage where they are active.** **a**. Heatmap showing the epigenetic status of MER11 elements – this is a replicate from Fig. 3d to allow visual reference of clusters that gain H3K27ac at different points of the differentiation. **b**. Per cluster breakdown of subfamily of MER11 elements. Some trends can be observed, such as MER11A elements being found in increased proportion in cluster (#1, 5 and #6) which show H3K27ac signal gain at S4. **c**. Heatmap showing the percentage of intersection with ChIP–seq of selected transcription factors and their relationship with either all MER11 elements (top) or epigenetic active clusters identified in Fig. 3c. **d**. Similarly, here is shown the intersection of activated clusters with KZFPs found to be enriched in MER11 elements.

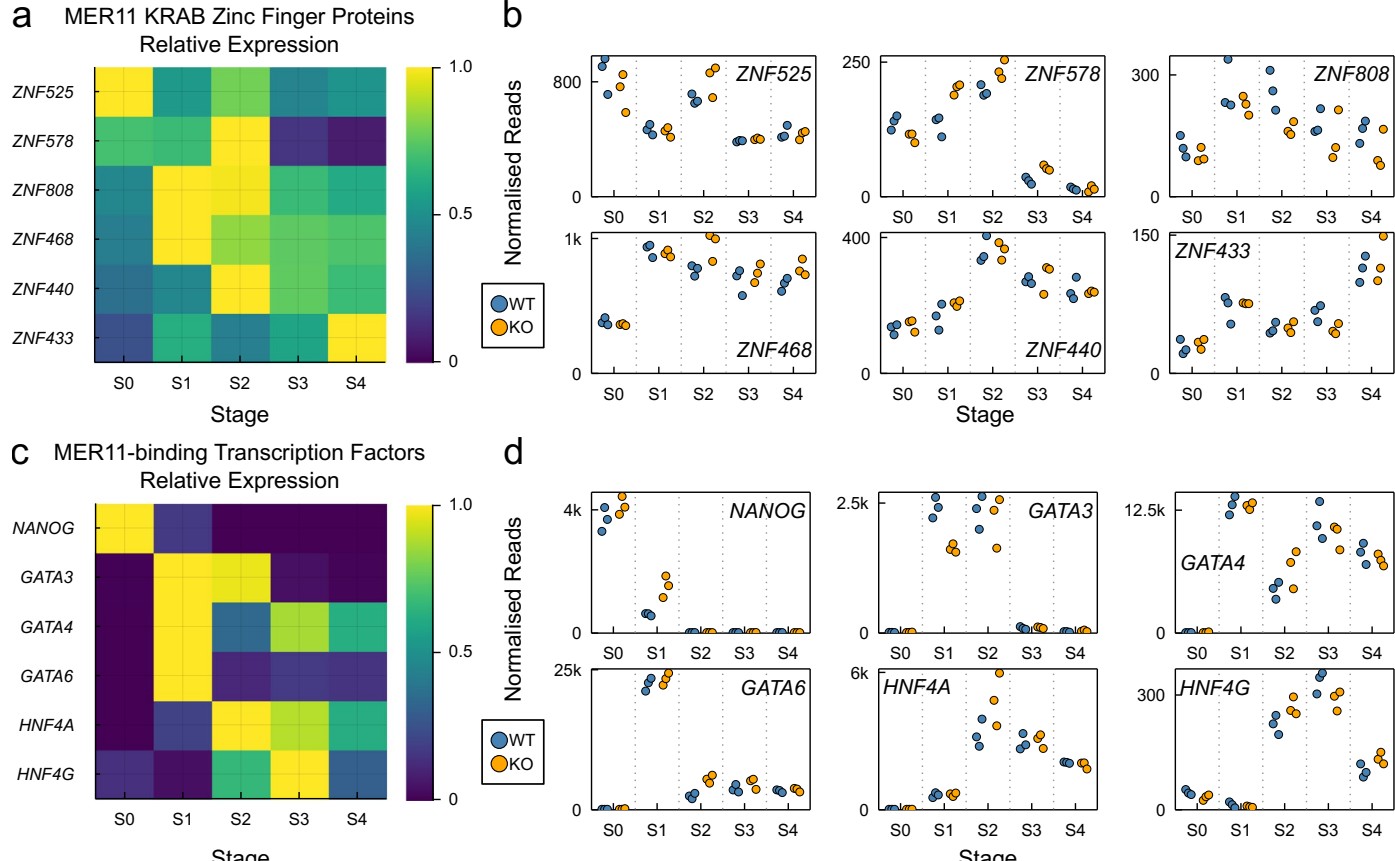

**Extended Data Fig. 6 | KZFPs and transcription factors binding MER11 elements have dynamic expression profiles during differentiation. a, b.** Expression of MER11-binding KZFPs. **a**- Heatmap showing maximum normalized mean expression in wild-type cells. **b**. Expression of all replicates in wild type and *ZNF808* KO. **c**–**d**. Expression of MER11-binding transcription factors. As (**a-b**) for transcription factors *GATA3, GATA4, GATA6, HNF4A, HNF4G*.

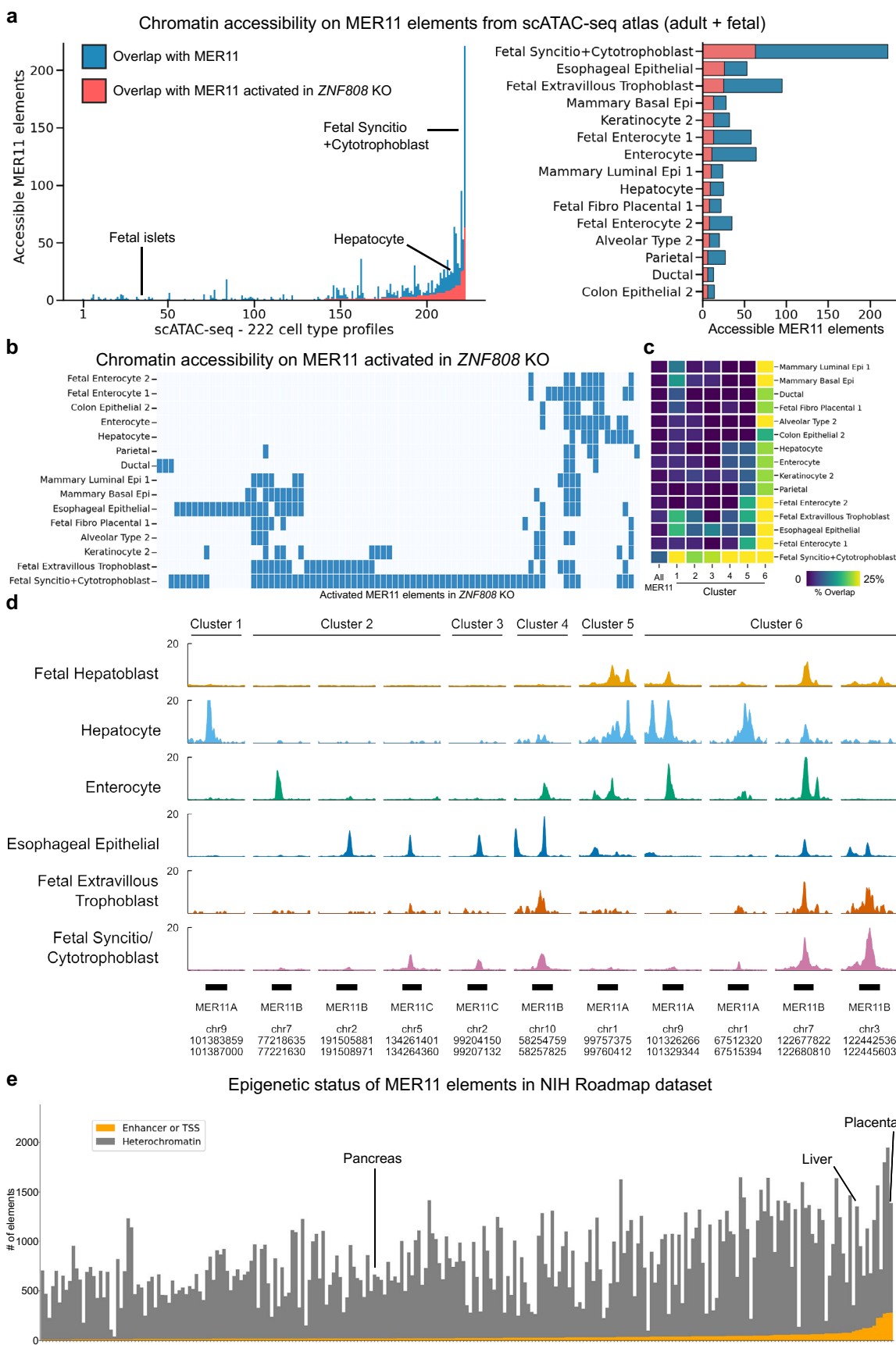

**Extended Data Fig. 7 | See next page for caption.**

**Extended Data Fig. 7 | MER11 elements are active in various cell types. a**. Overlap between accessible regions in 222 cell types from scATAC-seq human and fetal atlas and either all MER11 elements (blue) or the 220 elements that lose repression and gain activity in the *ZNF808* KO (red). On the right, a zoom of the 15 cell types with the highest overlap in the *ZNF808* KO activated MER11 elements. **b**. Heatmap of all MER11 elements activated in the *ZNF808* KO intersecting with a peak of chromatin accessibility in the scATAC-seq dataset. **c**. Percentage of overlap with scATAC-seq peaks in select cell types and clusters of H3K27ac activity found in the *ZNF808* KO during differentiation. Fetal syncytio/cytotrophoblasts are found in all 6 clusters at a higher percentage compared to MER11 background while other cell types are more enriched in cluster #6, which gains H3K27ac at S4. **d**. Examples of MER11 elements active in selected cellular contexts. All examples are taken from the list of MER11 losing H3K9me3 and gaining H3K27ac in the *ZNF808* KO. **e**. Epigenetic status of MER11 elements in the NIH Roadmap dataset – biosamples were collapsed per cell type and chromatin state predictions at the single base pair level were used. Here we focus on two chromatin states classes we have built from the aggregate of multiple smaller ones, H3K9me3-positive heterochromatin (containing H3K9me3-associated categories "ZNF_Rpts" and "Het") or Enhancer/TSS (aggregate of TSS-associated categories such as "TssA", "TssFlnk", "TssFlnkU", "TssFlnkD" and enhancer-associated categories such as "EnhG1", "EnhG2", "EnhA1", "EnhA2", "EnhWk"). There are 18 active MER11 elements for 'Pancreas' versus 100 for 'Liver'. Results show the same trends observed by ATAC-seq, notably that multiple MER11 elements are active in placenta or liver, but not in pancreas.

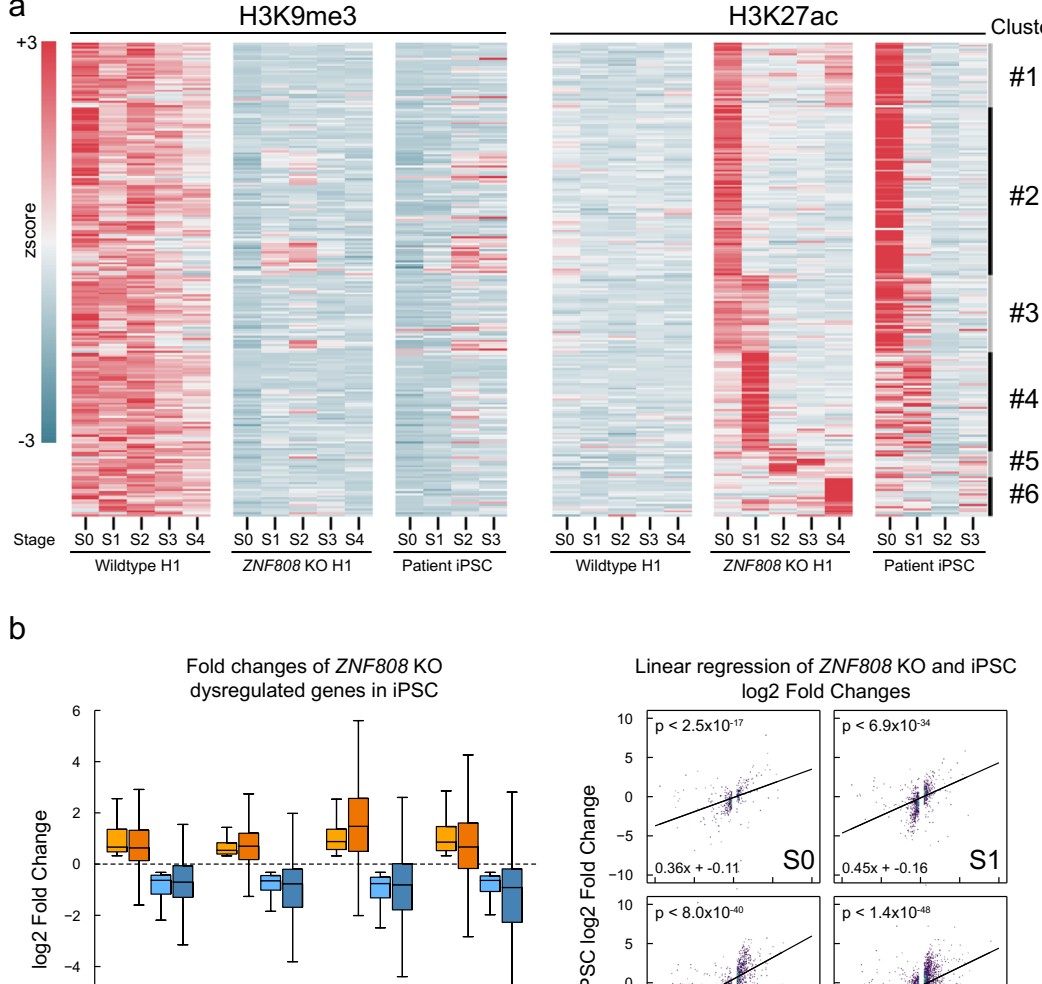

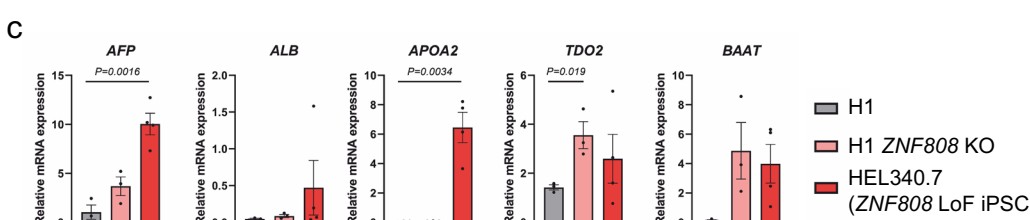

**Extended Data Fig. 8 | Validation of epigenetic and transcriptomic dysregulation observed in the *ZNF808* KO cells using patient-derived iPSCs.** **a**. Heatmap of the 220 MER11 elements that lose H3K9me3 and gain H3K27ac in the *ZNF808* KO as presented in Fig. 3, with the addition of signal obtained when differentiating iPSCs up to S3. **b**. Left, boxplots showing fold-change of activated and repressed dysregulated genes identified in *ZNF808* KO and patient-derived iPSCs, showing agreement in direction and magnitude of the gene expression perturbation. Boxes mark interquartile range, with central line describing the median and whiskers 1.5x interquartile range. N = total activated and repressed dysregulated genes, see Fig. 4a. Right, linear regressions between *ZNF808* KO and iPSC log2 fold-change KO over WT at same set of genes. Regression coefficients and p-value of slope term given. **c**. qRT-PCR for five hepatic marker genes assayed at the posterior foregut stage (S3) in cells derived from H1 control, H1-*ZNF808*-KO and the patient iPSC (line HEL340.7) carrying the *ZNF808* deletion (n = 3-4 independent differentiation experiments). Data are presented as mean values ± SEM. Unpaired two-tailed t test.

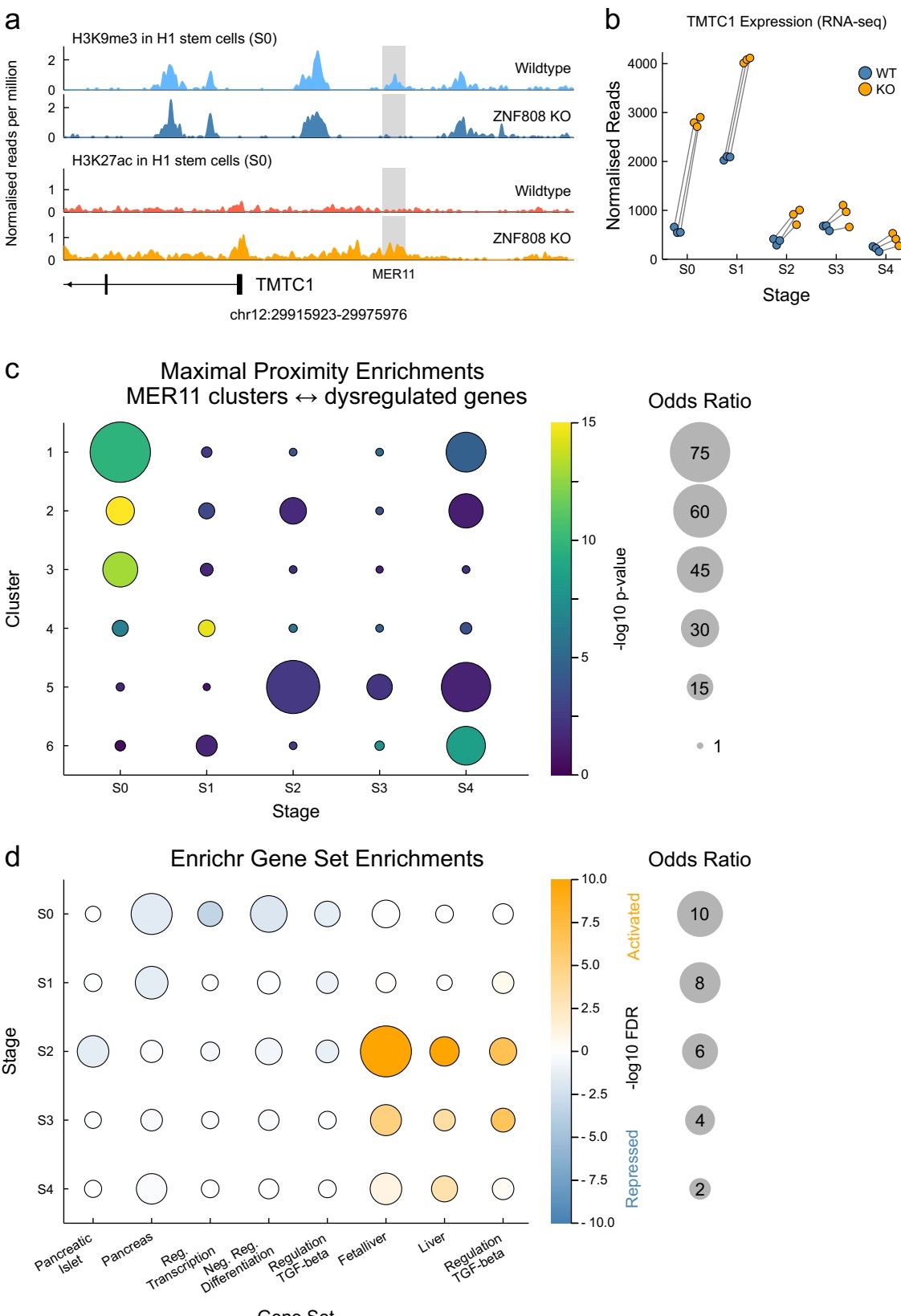

**Extended Data Fig. 9 | See next page for caption.**

**Extended Data Fig. 9 | Unmasked MER11 elements drive proximal gene activation in *ZNF808* KO and dysregulated genes are associated with a loss of pancreatic identity and a gain of hepatic identity. a**. Example locus showing loss of repression and activation of a MER11 element in the *ZNF808* KO with activation of the adjacent gene. Locus shows H3K9me3 and H3K27ac data in reads per million upstream of the *TMTC1* promoter. H3K9me3-marked MER11 element in wild type is lost at S0 in *ZNF808* KO concomitant with gain of H3K27ac. **b**. Gene expression for *TMTC1* shown in transcripts per million (TPM), showing robust upregulation at S0, S1 and S2 stages. **c**. Proximity enrichments between pairs of MER11 clusters identified in Fig. 3 and genes activated at each stage. Fisher exact test right-tail –log10 p-value is denoted by color and odds ratio by size of dot. Minimal p-value for gene-element pairs in the range [1 bp, 100 Mb] and the odds ratio at that minimum given. **d**. Enrichr gene set enrichments for activated (orange) or repressed (blue) genes in *ZNF808* KO selected gene sets and terms (see Supplementary Table 8) shown. Enrichments for activated genes are given for fetal liver, liver and regulation of TGF-beta, repressed otherwise. Fisher exact test right-tail –log10 p-value is denoted by color and odds ratio by size of dot.

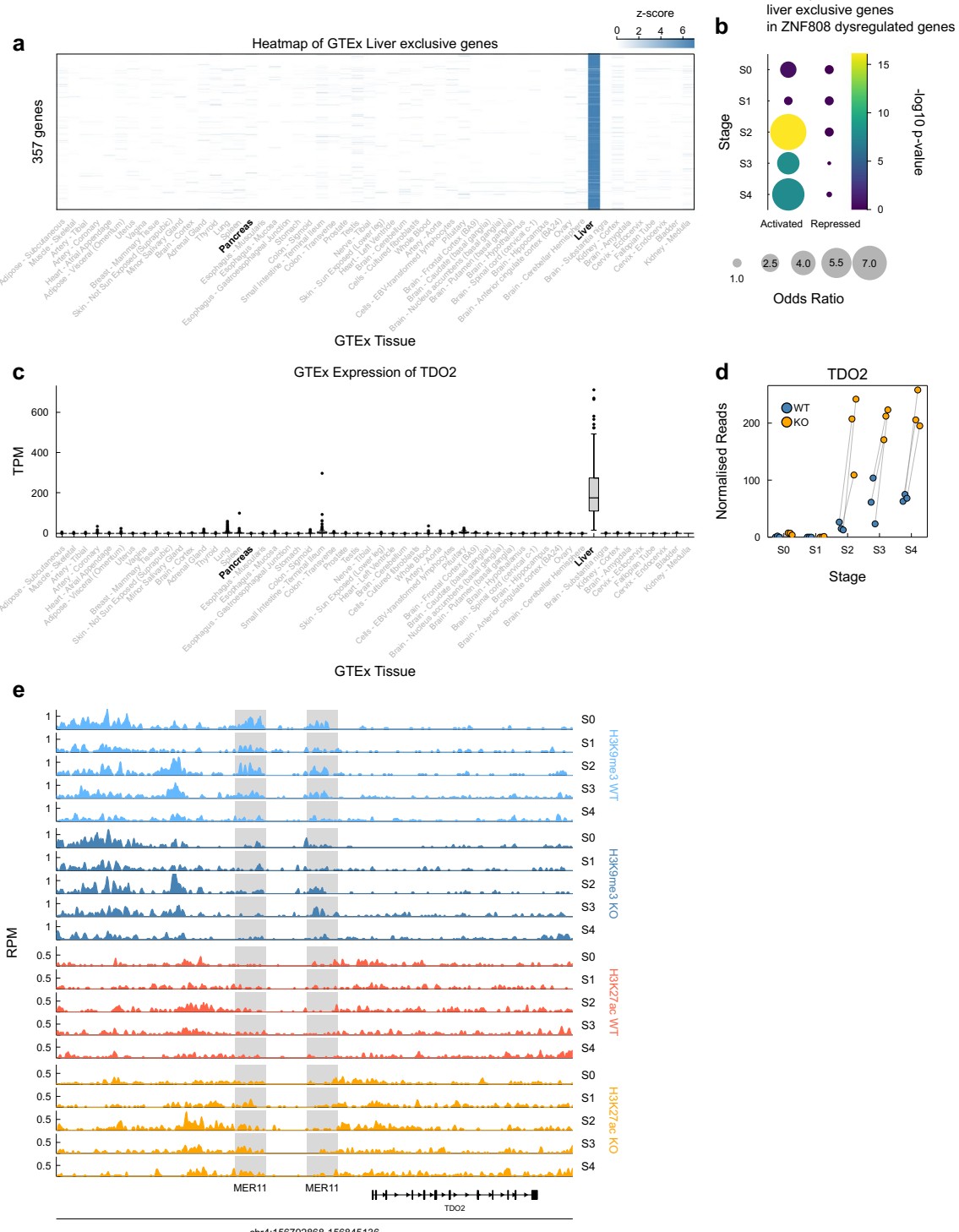

**Extended Data Fig. 10 | Genes exclusively expressed in liver are activated in *ZNF808* KO. a**. Heatmap of GTEx liver-exclusive genes. 357 genes for which their lower quartile of expression in liver exceeds the upper quartile of expression in all other tissues. Shown as z-score of median expression. Liver and pancreas highlighted. **b**. Over-representation of liver-exclusive genes in the list of genes activated in *ZNF808* KO. Bubble plot with Fisher Exact test -log10 p-value denoted by color and odds ratio denoted by dot size. **c-e**. *TDO2* is a liver-exclusive gene (**c**), activated at S2, S3 and S4 in *ZNF808* KO (**d**). *TDO2* is adjacent to two MER11 elements for which the leftmost loses H3K9me3 throughout the time course and gains H3K27ac from S1 onwards. Boxplots in (c) describe gene level *TDO2*

expression in the GTEx (Genotype-Tissue Expression) project. Data shows sum of GTEx v8 isoform level transcripts per million data calculated with RSEM. Boxplot central line denotes median, box limit the interquartile range and whiskers extend to furthest point within 1.5 interquartile range, data points are outliers exceeding whiskers, median sample size n = 291. The GTEx Project was supported by the Common Fund of the Office of the Director of the National Institutes of Health and by NCI, NHGRI, NHLBI, NIDA, NIMH and NINDS. The data used for the analyses described in this manuscript were obtained from the GTEx Portal on 08/26/21.

# Reporting Summary

## Statistics

For all statistical analyses, confirm that the following items are present in the figure legend, table legend, main text, or Methods section.

| n/a | Confirmed | |
|---|---|---|
| ☐ | ☒ | The exact sample size (*n*) for each experimental group/condition, given as a discrete number and unit of measurement |
| ☐ | ☒ | A statement on whether measurements were taken from distinct samples or whether the same sample was measured repeatedly |
| ☐ | ☒ | The statistical test(s) used AND whether they are one- or two-sided *Only common tests should be described solely by name; describe more complex techniques in the Methods section.* |
| ☒ | ☐ | A description of all covariates tested |
| ☐ | ☒ | A description of any assumptions or corrections, such as tests of normality and adjustment for multiple comparisons |
| ☐ | ☒ | A full description of the statistical parameters including central tendency (e.g. means) or other basic estimates (e.g. regression coefficient) AND variation (e.g. standard deviation) or associated estimates of uncertainty (e.g. confidence intervals) |
| ☐ | ☒ | For null hypothesis testing, the test statistic (e.g. *F*, *t*, *r*) with confidence intervals, effect sizes, degrees of freedom and *P* value noted *Give P values as exact values whenever suitable.* |
| ☒ | ☐ | For Bayesian analysis, information on the choice of priors and Markov chain Monte Carlo settings |
| ☒ | ☐ | For hierarchical and complex designs, identification of the appropriate level for tests and full reporting of outcomes |
| ☐ | ☒ | Estimates of effect sizes (e.g. Cohen's *d*, Pearson's *r*), indicating how they were calculated |

*Our web collection on statistics for biologists contains articles on many of the points above.*

## Software and code

Policy information about availability of computer code

| Data collection | No software used to collect data |
|---|---|
| Data analysis | GATK v3.7 Haplotypecaller, AlamutBatch v1.8, SavvyCNV, SavvyVcfHomozygosity, Benchling, Geneious Prime 2020.1.1, CRISPOR, FlowJo v9, Bowtie2 version 2.4.4, MACS2 2.2.7.1, bedtools v2.30.0, STAR v2.7.3a, RSEM v1.3.2, DESeq2 v1.30.1, ProximityEnrichment.jl, Enrichr, MAFFT v7.475, pybedtools 0.8.1, samtools 1.18, HTSeq 0.13.5, pandas 1.3.5, numpy 1.21.5, scipy 1.7.3, fastcluster 1.2.4, BD CellQuest Pro v4.0.2 , Rotor-Gene Q. Source data and code is available from https://github.com/owensnick/ZNF808Genomics.jl and https://doi.org/10.5281/zenodo.8375708 |

For manuscripts utilizing custom algorithms or software that are central to the research but not yet described in published literature, software must be made available to editors and reviewers. We strongly encourage code deposition in a community repository (e.g. GitHub). See the Nature Portfolio guidelines for submitting code & software for further information.

## Data

Policy information about availability of data

All manuscripts must include a data availability statement. This statement should provide the following information, where applicable:

- Accession codes, unique identifiers, or web links for publicly available datasets
- A description of any restrictions on data availability
- For clinical datasets or third party data, please ensure that the statement adheres to our policy

Transcriptomic and Epigenomic data are available from NCBI GEO under accession GSE205164. Clinical and genotype data is available only through collaboration as this can be used to identify individuals and so cannot be made openly available. Requests for collaboration will be considered following an application to the Genetic Beta Cell Research Bank (https://www.diabetesgenes.org/current-research/genetic-beta-cell-research-bank/). Contact by email should be directed to Prof Andrew Hattersley.

## Human research participants

Policy information about studies involving human research participants and Sex and Gender in Research.

| | |
|---|---|
| Reporting on sex and gender | We did not use the term gender in our report and the genetic findings apply to both sexes. Sex information was collected from the referring clinicians and confirmed by the genetic analysis. Recruitment in the study was based on diagnosis of neonatal diabetes, which is a rare disease characterised by diabetes onset in the first six months of life. Anyone with the diagnosis was recruited in the study, independently of sex. |
| Population characteristics | The characteristics of patients homozygous for ZNF808 mutations are summarized in Supplementary Table 3 |
| Recruitment | Recruitment in the study was based on diagnosis of neonatal diabetes, which is a rare disease characterised by diabetes onset in the first six months of life. Individuals were referred by their clinicians for genetic testing of neonatal diabetes to the Exeter Genomics Laboratory. Genetic testing for neonatal diabetes was provided free of charge as it was covered by research funds. We are not aware of any bias in patients' recruitment for our study. |
| Ethics oversight | The study was conducted in accordance with the Declaration of Helsinki, and all subjects or their parents/guardian gave informed consent for genetic testing. Wales Research Ethics Committee 5 Bangor (REC 17/WA/0327). |

Note that full information on the approval of the study protocol must also be provided in the manuscript.

# Field-specific reporting

Please select the one below that is the best fit for your research. If you are not sure, read the appropriate sections before making your selection.

☒ Life sciences          ☐ Behavioural & social sciences          ☐ Ecological, evolutionary & environmental sciences

For a reference copy of the document with all sections, see nature.com/documents/nr-reporting-summary-flat.pdf

# Life sciences study design

All studies must disclose on these points even when the disclosure is negative.

| | |
|---|---|
| Sample size | No statistical method was used to determine the genetic testing cohort's size as our study aimed at identifying novel genetic causes of pancreatic agenesis, which is a rare monogenic condition.<br>For transcriptomic and epigenomic surveying cells over in vitro differentiation time course standard sample sizes within the field were used. For transcriptomic data n = 3 for wt and ZNF808 KO and n = 1 for the patient-derived iPSC. From each of wildtype, ZNF808 KO and patient-derived iPSC one replicate per stage was assayed for H3K9me3 and H3K27ac epigenomics. |
| Data exclusions | For transcriptomic and epigenomic data genes and peaks respectively were excluded with low read counts. |
| Replication | Three independent replicates of H1 embryonic stem cell derived wild type and ZNF808 KO differentiation timecourses were collected. all attempts at replication were successful. |
| Randomization | Our study design did not include randomization as we investigated a group of patients with neonatal diabetes - there was no allocation of individuals to different groups. |
| Blinding | Since individuals were not allocated to different groups, blinding was not applicable to our study. |

# Reporting for specific materials, systems and methods

We require information from authors about some types of materials, experimental systems and methods used in many studies. Here, indicate whether each material, system or method listed is relevant to your study. If you are not sure if a list item applies to your research, read the appropriate section before selecting a response.

## Materials & experimental systems

| n/a | Involved in the study |
|---|---|
| ☐ | ☒ Antibodies |
| ☐ | ☒ Eukaryotic cell lines |
| ☒ | ☐ Palaeontology and archaeology |
| ☒ | ☐ Animals and other organisms |
| ☒ | ☐ Clinical data |
| ☒ | ☐ Dual use research of concern |

## Methods

| n/a | Involved in the study |
|---|---|
| ☐ | ☒ ChIP-seq |
| ☐ | ☒ Flow cytometry |
| ☒ | ☐ MRI-based neuroimaging |

## Antibodies

| Antibodies used | Histone H3K27ac antibody (mAb) - Catalog number 39685, Active Motif; Histone H3K9me3 antibody (pAb) - Catalog number 39161, Active Motif<br>Mouse Anti-CD184 (CXCR4), PhycoErythrin Conjugated, 1:10 (FC), BD Biosciences Cat# 555974<br>Mouse IgG2a, PhycoErythrin Conjugated, 1:10 (FC), BD Biosciences Cat# 563023<br>Mouse Anti-PDX1, Phycoerythrin Conjugated, 1:80 (FC), BD Biosciences Cat# 562161<br>Mouse Anti-NKX6-1, Alexa Fluor 647 Conjugated, 1:80 (FC),  BD Biosciences Cat# 563338<br>Rabbit IgG Isotype Control, Alexa Fluor 647 Conjugate, 1:80 (FC), Cell Signaling Technology Cat# 3452S<br>Goat anti-PDX1, 1:250 (ICC), R and D Systems Cat# AF2419<br>Mouse anti-NKX6-1, 1:250 (ICC), DSHB Cat# F55A10<br>Rabbit anti-AlphaFetoProtein, 1:500 (ICC), Dako Cat# A0008<br>Rabbit anti-SOX9, 1:250 (ICC), Millipore Cat# AB5535<br>Sheep anti-NEUROG3, 1:250 (ICC), R and D Systems Cat# AF3444<br>Donkey anti-Goat conjugate, Alexa Fluor 488 1:500 (ICC), Thermo Fisher Scientific Cat# A-11055<br>Donkey anti-Rabbit conjugate, Alexa Fluor 594 1:500 (ICC), Thermo Fisher Scientific Cat# A-21207<br>Donkey anti-Mouse conjugate, Alexa Fluor 488  1:500 (IcC), Thermo Fisher Scientific Cat# A-21202<br>Donkey anti-Sheep conjugate, Alexa Fluor 594 1:500 (ICC), Thermo Fisher Scientific Cat# A-11016 |
|---|---|
| Validation | ChIP-Seq validation was performed by Active Motif's Epigenetics Services; the complete data set is available on their website. On our end, we compared results with expected profiles for these epigenetic marks at positive control loci - gene promoters for H3K27ac, and 3' end of zinc finger genes for H3K9me3.<br>All antibodies used were validated using primary islet tissue or in-house tissue samples. The anti-CD184-PE, anti-PDX1-PE, and anti-NKX6-1-AF647 antibodies have flow cytometry validation data and relevant citations on the BD Biosciences website. The anti-PDX1 and anti-NGN3 antibodies have IF validation images on the R and D Systems website as well as numerous relevant citations. The anti-NKX6-1 antibody has IF validation images on DSHB website as well as numerous relevant citations. The anti-SOX9 antibody has IF validation images on Merck website as well as numerous relevant citations. The anti-AFP antibody has been developed against Alpha-1-fetoprotein isolated from human cord serum and verified by the provider using crossed immunoelectrophoresis and ELISA, in addition to several relevant citations. The anti-goat AF488, anti-rabbit AF594, anti-sheep AF594, and anti-mouse AF488 antibodies all have numerous relevant citations of validated function on the product website (Thermo Fisher). |

## Eukaryotic cell lines

Policy information about cell lines and Sex and Gender in Research

| Cell line source(s) | H1 hESCs were purchased from WiCell. H1-ZNF808-/- were generated in Timo Otonkoski's lab using CRISPR/Cpf1. HEL 214.10 and HEL 340 were generated at the Biomedicum Stem Cell Center, University of Helsinki from donated skin fibroblasts under the approval of the Coordinating Ethics Committee of the Helsinki and Uusimaa Hospital District (no. 423/13/03/00/08) |
|---|---|
| Authentication | H1, H1-ZNF808-/-, HEL214 and HEL340 were validated with methods including G-band karyotyping and quantitative PCR for pluripotency markers (OCT4, NANOG, SOX2). |
| Mycoplasma contamination | The hPSCs were tested routinely for Mycoplasma and they tested negative |
| Commonly misidentified lines (See ICLAC register) | Not used |

# ChIP-seq

## Data deposition

☒ Confirm that both raw and final processed data have been deposited in a public database such as GEO.

☒ Confirm that you have deposited or provided access to graph files (e.g. BED files) for the called peaks.

Data access links
*May remain private before publication.*

https://www.ncbi.nlm.nih.gov/geo/query/acc.cgi?acc=GSE205164
Reviewer access token: qxmteweandyftqx

Files in database submission

Broadpeak files generated with MACS2 for all H3K9me3 and H3K27ac ChIP-seq for differentiation stages S0-S4 in WT and ZNF808 KO.

Genome browser session
(e.g. UCSC)

NA

## Methodology

Replicates

1 replicate of each of H3K9me3 and H3K27ac H1 WT and ZNF808 KO for stages S0-S4, and 1 replicate of HEL214 for stages S0-S3

Sequencing depth

Each experiment had at least 20M 150 bp paired-end reads (40M total reads), with some experiments reaching 30M paired-end reads (60M total reads).

Antibodies

Histone H3K27ac antibody (mAb) - Catalog number 39685, Active Motif; Histone H3K9me3 antibody (pAb) - Catalog number 39161, Active Motif

Peak calling parameters

Read mapping with Bowtie2 2.4.4
bowtie2 --very-sensitive-local -x index -1 chip_R1.fq.gz -2 chip_R2.fq.gz
Peak calling:
macs2  callpeak -g hs -c input.bam -t chip.bam  --broad

Data quality

H1_S0_H3K9me3.broadPeak  65929
H1_ZNF808KO_S0_H3K9me3.broadPeak  75117
H1_S1_H3K9me3.broadPeak  72607
H1_ZNF808KO_S1_H3K9me3.broadPeak  84600
H1_S2_H3K9me3.broadPeak  72024
H1_ZNF808KO_S2_H3K9me3.broadPeak  66170
H1_S3_H3K9me3.broadPeak  57763
H1_ZNF808KO_S3_H3K9me3.broadPeak  57487
H1_S4_H3K9me3.broadPeak  46494
H1_ZNF808KO_S4_H3K9me3.broadPeak  42307

H1_S0_H3K27ac.broadPeak  37006
H1_ZNF808KO_S0_H3K27ac.broadPeak  74265
H1_S1_H3K27ac.broadPeak  46609
H1_ZNF808KO_S1_H3K27ac.broadPeak  45016
H1_S2_H3K27ac.broadPeak  35596
H1_ZNF808KO_S2_H3K27ac.broadPeak  41989
H1_S3_H3K27ac.broadPeak  17521
H1_ZNF808KO_S3_H3K27ac.broadPeak  28633
H1_S4_H3K27ac.broadPeak  31481

Software

Read mapping: Bowtie2 2.4.4
Peak Calling: MACS2 2.2.7.1
Peak Intersection: bedtools v2.30.0

# Flow Cytometry

## Plots

Confirm that:

☒ The axis labels state the marker and fluorochrome used (e.g. CD4-FITC).

☒ The axis scales are clearly visible. Include numbers along axes only for bottom left plot of group (a 'group' is an analysis of identical markers).

☒ All plots are contour plots with outliers or pseudocolor plots.

☒ A numerical value for number of cells or percentage (with statistics) is provided.

## Methodology

**Sample preparation**

hPSC-derived cells were dissociated with TryplE for 5-10 minutes at 37°C before fixation/permeabilisation in BD Cytofix/CytoPerm solution for 20 minutes at 4°C. Primary conjugated antibodies were incubated for 30 min at room temperature or overnight at 4°C in a 5% FBS/PBS solution.

**Instrument**

FACSCalibur cytometer (BD Bioscience)

**Software**

For Acquisition: BD CellQuest Pro v4.0.2
For analysis: FlowJo software v9 (Tree Star Inc.)

**Cell population abundance**

The major endocrine cell populations were in high prevalence in this study. Cell population abundances ranged from 40% to >90%

**Gating strategy**

Cells were gated with FSC and SSC to remove small interfering cellular debris. Positive and negative gating was determined through negatively stained cells within the population and non-stained controls.

☒ Tick this box to confirm that a figure exemplifying the gating strategy is provided in the Supplementary Information.

