## [Peer Review File · Nature Genetics]

Peer Review Information

Manuscript Title: Primate-specific ZNF808 is essential for pancreatic development in humans

Corresponding author name(s): Professor Andrew (T) Hattersley, Professor Timo Otonkoski, Dr Michael Imbeault

Reviewer Comments & Decisions:

Decision Letter, initial version:
--

30th Mar 2023

Dear Professor Hattersley,

Your Letter, "Primate-specific ZNF808 is essential for pancreatic development in humans" has now been seen by 3 referees. You will see from their comments below that while they find your work of interest, some important points are raised. We are interested in the possibility of publishing your study in Nature Genetics, but would like to consider your response to these concerns in the form of a revised manuscript before we make a final decision on publication.

We therefore invite you to revise your manuscript taking into account all reviewer and editor comments, and addressing them in full. Please highlight all changes in the manuscript text file. At this stage we will need you to upload a copy of the manuscript in MS Word .docx or similar editable format.

*2) If you have not done so already please begin to revise your manuscript so that it conforms to our Letter format instructions, available [here](http://www.nature.com/ng/authors/article_types/index.html).

*3) Include a revised version of any required Reporting Summary:

[redacted]

We hope to receive your revised manuscript within four to eight weeks. If you cannot send it within this time, please let us know.

Sincerely,

Safia Danovi
Editor
Nature Genetics

Referee expertise

Referee #1: KRAB, TEs

Referee #2: TEs in development (signed report)

Referee #3: pancreatic development and diabetes

Reviewers' Comments:

Reviewer #1:

Remarks to the Author:

An interesting study demonstrating the implication of the primate-specific ZNF808 in the development of the human pancreas via the stage-specific repression of Mer11-regulated liver-promoting genes.

Specific comments:

1. On the figures / data:

- Fig. 2 and 3 have the same title.
- Fig. 2a legend, reference to track evolution of ZNF808 must be 6, not 5.
- Fig. 3c left panel would benefit from statistics, and a better description of what is depicted.
- Fig. 4b: random genes should be used as a control.
- Fig. 4d: are liver-exclusive genes closer to Mer11 inserts? To ZNF808-recruiting ones?

2. On the text:

- not sure the first sentence is the best way to introduce the topic. It may happen to be true here, but how generalizable is this statement? Are there many other examples?
- a presentation of pancreatic agenesis (incidence, known causing variants, ...) in the introduction would help appreciate what follows, rather than having to wait for the discussion to get an overview of the disease.
- is there a behind-the-curtain explanation for the initial finding of 2 homozygous ZNF808 l.o.f. in 2 cases of pancreatic agenesis, and then only 13 out of an additional 233? Was there a bias in the initial exploration?

3. Additional questions:

- What about the other KZFPs binding to Mer11? Are there suspected pathogenic variants in those? What is their pattern of expression and potentially differential recruitment on Mer11 units linked to a liver-directed transcriptional program?
- Are Mer11 inserts found here to be ZNF808-repressed during pancreatic development acetylated in the adult liver? What about the chromatin status of these loci in the adult pancreas?
- Reference 6 and Suppl. Fig 9b defined ZNF808-binding and non-binding subsets within Mer11 inserts, with notably a group of Mer11c apparently escaping this KZFP. Does it correlate with differential distribution / expression of their neighboring genes or distinct patterns of transcription factors recruitment (not visible in Suppl. Fig. 9b)?
- The model would be consolidated by verifying that KO cells can be complemented by overexpression of ZNF508 or CRISPRi-mediated repression of Mer11 in the in vitro pancreatic differentiation system.

Reviewer #2:

Remarks to the Author:

The manuscript by De Franco, Owens, Montaser et al. shows that ZNF808 loss-of-function mutations cause pancreatic agenesis. Using an hESC differentiation model, they demonstrate that ZNF808 prevents undue expression of liver-specific genes by silencing MER11 TEs that putatively act as enhancers when active.

This is a really surprising and incredibly interesting finding. The assumption has always been that recently evolved proteins should play relatively minor (and/or redundant) roles in basic processes such as development and organogenesis. Similar arguments could be made about TEs, although the paradigm is starting to shift on this matter. This paper certainly contributes to that shift, uncovering a firm example of how the KZPF-TE relationship has shaped human evolution.

The study is generally well conducted, with solid genetic data and extensive analysis of the in vitro KO model. My main comment for improvement is related to the fact that all hESC results appear to have been derived from a single clone. Although the data presented from this clone are convincing, it would be reassuring to see key results replicated in multiple null CRISPR clones (with an equal number of wildtype clones derived during CRISPR). I am assuming here that obtaining a null clone wasn't particularly difficult, and that multiple null clones were stored, which might not be the case. Alternatively, or complementarily, if patient-derived iPSCs were available, this would also add robustness to the study.

My only other comment is minor: it would be useful to provide the number of elements of each MER11 subfamily within each of the clusters in Figure 3c. Mainly because I think this will point to a major role of MER11A in driving liver-specific gene expression. This is the subfamily that bears the relevant motifs and binds HNF4A, for example. Also, in Supp. Fig. 8d, virtually every example of hepatocyte-active elements comes from the MER11A subfamily.

Miguel Branco

Reviewer #3:

Remarks to the Author:

Summary of the key results

The manuscript "Primate-specific ZNF808 is essential for pancreatic development in humans" by De Franco et al. identifies a novel source for congenital pancreas agenesis – mutation in the DNA binding gene ZNF808. Notably, this gene is primate-specific, and as such this is "the first report of loss of a primate-specific gene causing a congenital developmental disease". The authors move on to uncover the mechanism underlying the pancreas's lack of ability to form, and place the finger on ZNF808's role as a suppressor of the MER11 family of transposable elements. In its absence, the elements that are normally silenced become activated, and their transcription dysregulates the balance between the pancreatic and the hepatic gene expression programs.

The identification of a novel mutation responsible for pancreas agenesis is very interesting, and it being in a gene unique to primates further increases the interest of the work. In addition, the gene's mechanism of action – as presented by the authors – is fascinating. However, as detailed below, I believe that several important claims made by the authors are not adequately supported by their findings. Briefly, the manuscript can be divided into two parts. In the first, the authors nicely show that mutations in ZNF808 can be linked to pancreas agenesis. This claim is supported by highly convincing data. In the second part of the manuscript, the authors describe ZNF808's mechanism of action. The evidence in this part is lacking, and in fact brings to questions why it's known effect is limited to the pancreas.

Major concerns

1) To prove ZNF808's mechanism of action, the authors use an in vitro protocol for the directed

differentiation of pluripotent cells into pancreatic progenitors, and observe an impact of ZNF808 KO on the cells' epigenetic and transcriptional landscape:

- a. It is unclear if the authors see any effect of the gene's deletion on the quality of pancreas differentiation. If no effect is observed, the relevance of the in vitro model to the issue at hand can be questioned. Perhaps examining the effect of the gene's loss on later stages of differentiation (e.g. endocrine differentiation) may reveal a stronger phenotype.
- b. Pancreas agenesis includes malformation of both the acinar and the endocrine components of the pancreas, but the authors do not test a protocol for acinar differentiation. This may be excluded if the endocrine differentiation used would have shown a substantial phenotype.
- c. The most significant effect of ZNF808 KO on MER11 expression seems to be in S0, when cells are still in their undifferentiated state. However, there is no mention of a potential effect on the differentiation into tissues other than the pancreas – preferably non-endodermal tissues - to examine if this KO indeed affects only the pancreas.

2) Based on transcriptional analysis, the authors claim that “the unmasking of MER11 elements in the ZNF808 KO has direct impact on gene expression”, and present this impact as the underlying mechanism of ZNF808's mode of action. However, the manuscript does not show any direct evidence for an effect of MER11's transcription on gene expression. The data only shows correlation between the effect of ZNF808's KO on MER11's expression and on the expression of the dysregulated genes. It could be, for example, that ZNF808 is directly responsible for both the gene's change in expression and, independently, on MER11 expression.

Additional issues

- 1) supplementary figure 6 and Figure 2c are difficult to interpret.
- 2) In supplementary Figure 1b – in situ hybridization of ZNF808 during pancreas development – the staining is not convincing. It is hard to see in which cells it is expressed. In addition, a non-pancreatic control sample could be helpful, to show that staining is indeed specific.
- 3) In figure 1b – can a statistical value be assigned to the likelihood of the findings? i.e. what is the statistical significance of the observed pattern of inheritance?

Author Rebuttal to Initial comments

We are grateful for the reviewers' enthusiastic comments and constructive feedback. We have addressed all the points raised in the document below, including new data, additional analysis, and clarifications in the text. We would be happy to include any of the new data provided into the manuscript if recommended by the reviewers and the editors.

Reviewer #1:

Remarks to the Author:

An interesting study demonstrating the implication of the primate-specific ZNF808 in the development of the human pancreas via the stage-specific repression of Mer11-regulated liver-promoting genes.

Specific comments:

1. On the figures / data:

1a. Fig. 2 and 3 have the same title.

We apologize for this formatting mistake; this has now been corrected.

1b. Fig. 2a legend, reference to track evolution of ZNF808 must be 6, not 5.

We thank the reviewer for spotting this inconsistency, it has now been corrected (now reference 8).

1c. Fig. 3c left panel would benefit from statistics, and a better description of what is depicted.

We have now performed Fisher's Exact tests to determine if MER11 elements known to be bound by ZNF808 are significantly more likely to lose H3K9me3 or gain H3K27ac than those that are not known to be bound by ZNF808. At the aggregate level (if a MER11 lost / gained epigenetic marks in at least 1 stage of differentiation) both are highly significant - H3K9me3 loss with $p < 3.6 \times 10^{-61}$ and H3K27ac gain with $p < 1.8 \times 10^{-16}$. We have edited the manuscript to include this. We have also split the Figure 3c panel in two (c and d) to add clarity and updated the legend and text accordingly.

1d. Fig. 4b: random genes should be used as a control.

We used the proximity of all genes to MER11 elements as a control in Figure 4b which depicts Fisher exact test $-\log_{10}$ p-values for the association between a gene being dysregulated and being proximal to a MER11 element. We have updated the figure legend to clarify this point. As we are sure the reviewer agrees, all genes is a superior control to a random subset which would provide a similar result but with added sampling noise.

1e. Fig. 4d: are liver-exclusive genes closer to Mer11 inserts? To ZNF808-recruiting ones?

In our study we found that MER11 elements activation drive early gene expression differences, and that the induction of liver-exclusive genes is largely downstream of this. To demonstrate this more clearly, we have now calculated the proximity enrichments between liver exclusive genes and differing sets of MER11 elements/ZNF808 peaks:

Figure legend – Proximity enrichments: $-\log_{10}$ Fisher exact test p-values for the association between liver-exclusive genes and genes being within given distance of a MER11 element of the given group, and we show genes activated at S0 as a control.

Overall, as a group, we do not find liver-exclusive genes enriched in proximity to MER11 elements. There are notable exceptions, such as TDO2 and BAAT, that are upregulated and located in close proximity to activated MER11 elements which we highlight in Supplementary Figure 10 panel d. We therefore find that MER11 activation is compatible with a liver-exclusive programme, but most genes identified as part of the liver-associated programme are not driven by MER11 elements directly.

2. On the text:

2a. not sure the first sentence is the best way to introduce the topic. It may happen to be true here, but how generalizable is this statement? Are there many other examples?

- a presentation of pancreatic agenesis (incidence, known causing variants, ...) in the introduction would help appreciate what follows, rather than having to wait for the discussion to get an overview of the disease.

We have now edited the first paragraph of the manuscript to address the reviewer's suggestions.

2b. is there a behind-the-curtain explanation for the initial finding of 2 homozygous ZNF808 l.o.f. in 2 cases of pancreatic agenesis, and then only 13 out of an additional 233? Was there a bias in the initial exploration?

The initial (discovery) and subsequent (replication) investigations were carried out in different cohorts, which explains the different proportion of cases explained by ZNF808 variants. As mentioned in the methods, the 2 patients analysed in the discovery phase were selected from a larger cohort (n=107) of individuals with a confirmed diagnosis of pancreatic agenesis (defined as neonatal diabetes and exocrine pancreatic insufficiency). Within this cohort there were only 6 individuals without a pathogenic variant in a known gene and patients 1 and 2 were the only two among them who were born to consanguineous parents, suggesting that a homozygous causative variant was likely and were therefore selected for this study. The second stage of the genetic study (the replication) was performed in an unselected cohort of individuals with neonatal diabetes for whom their exocrine function was unknown. There are >30 causative genes which explain 85% of the neonatal diabetes cases and the replication cohort was composed of individuals for whom these known causes had been excluded. The replication cohort is therefore both clinically and genetically more heterogeneous than the pancreatic agenesis discovery cohort, thereby explaining the different proportion of individuals with homozygous ZNF808 variants identified.

3. Additional questions:

3a. What about the other KZFPs binding to Mer11? Are there suspected pathogenic variants in those?

We did not identify any pathogenic variants in the 5 genes encoding the other KZFPs binding the MER11 elements (ZNF440, ZNF433, ZNF525, ZNF468, and ZNF578) in the 193 genome-sequenced samples from individuals with neonatal diabetes without a known genetic cause performed by the Exeter team.

Investigation of additional samples from patients with neonatal diabetes and other diseases will be

necessary to establish if variants in these genes are also a genetic cause of Mendelian disease(s) in humans.

3b. What is their pattern of expression and potentially differential recruitment on Mer11 units linked to a liver-directed transcriptional program?

The pattern of expression of other KZFPs that bind MER11 is shown in in Supplemental Figure 1 and 7:

- *Supplemental Figure 1 shows that all KZFPs are variably expressed between tissues during human development, with all being comparatively lower in embryonic liver when compared to embryonic pancreas.*
- *Supplemental Figure 7 shows that all KZFPs binding MER11s are dynamically expressed during in vitro pancreatic differentiation with various patterns observed – ZNF433 and ZNF468 increase during differentiation, ZNF578 decreases at later time points, whilst ZNF808 and ZNF440 peak at intermediary stages.*

Regarding the recruitment of other KZFPs in relationship with a liver-directed programme, we can point out as shown in Supplementary Figure 3 that ZNF440 binding correlates extremely well with endoderm and liver transcription factor HNF4a and GATA6 binding, which suggest a pattern of co-selection. This section of MER11 elements is specific to MER11A elements and contains multiple HNF4a binding motifs. We are planning further work to investigate this interesting question over the next few years.

3c. Are Mer11 inserts found here to be ZNF808-repressed during pancreatic development acetylated in the adult liver? What about the chromatin status of these loci in the adult pancreas?

We have indeed investigated this interesting point and our results are provided in Supplementary Figure 8 panel e of our manuscript. Briefly, this figure depicts the intersect between MER11 elements and epigenetic states derived from a large-scale dataset of adult tissues by the NIH Roadmap consortium. The data we used is a predicted epigenetic state call derived from an ensemble of epigenetic marks, including H3K27ac. For the analysis, we have grouped a subset of the 18 states they provide to only retain those related to H3K9me3 heterochromatin (which in the figure is labelled 'Heterochromatin') or Enhancer / TSS (a mix of H3K27ac, H3K4me1 and H3K4me3). We find 18 active MER11 elements for 'Pancreas' versus 100 for 'Liver'. The 220 unmasked MER11 in the ZNF808 KO follow a similar trend with 4 elements active in 'Pancreas' and 20 in 'Liver'. The conclusion is that in pancreas, MER11 elements are almost completely inactive with a few elements detected as TSS or enhancer, while adult liver is one of the tissue types where we find the highest number of MER11 elements in an active epigenetic state (TSS or enhancer).

3d. Reference 6 and Suppl. Fig 9b defined ZNF808-binding and non-binding subsets within Mer11 inserts, with notably a group of Mer11c apparently escaping this KZFP. Does it correlate with differential distribution / expression of their neighboring genes or distinct patterns of transcription factors recruitment (not visible in Suppl. Fig. 9b)?

It is our understanding that the reviewer is referring here to Supplementary Figure 5b. The group of MER11C that is escaping repression is bound primarily by ZNF525 and ZNF578, as shown in the multiple alignment of MER11 elements with overlaid signal of KZFPs in Supplementary Figure 3. It is apparent that the younger MER11C elements are repressed by the younger ZNF525 and ZNF578 and have mostly escaped ZNF808 control. The same figure shows a local enrichment on these sequences of DUX4 and SMAD2.

We do not yet know what the interplay is between the 6 KZFPs and transcription factors in regulation of domesticated platforms derived from MER11 elements – we hypothesize as discussed in the manuscript that this forms the basis of a combinatorial system of regulation, and we are planning to further investigate this hypothesis in the future.

3e. The model would be consolidated by verifying that KO cells can be complemented by overexpression of ZNF508 or CRISPRi-mediated repression of Mer11 in the in vitro pancreatic differentiation system. *While we agree that validation is important, we have chosen a different route than those suggested by the reviewer for practical reasons. We instead opted to provide validation through data obtained from a ZNF808 patient-derived iPSCs, shown below as a reply to point 1 raised by reviewer #2. Results agree and confirm patterns observed in our ZNF808 KO stem cell line derived using CRISPR, validating the effects we are describing in the manuscript are not due to off target effects of the CRISPR process.*

Reviewer #2:

Remarks to the Author:

The manuscript by De Franco, Owens, Montaser et al. shows that ZNF808 loss-of-function mutations cause pancreatic agenesis. Using an hESC differentiation model, they demonstrate that ZNF808 prevents undue expression of liver-specific genes by silencing MER11 TEs that putatively act as enhancers when active.

This is a really surprising and incredibly interesting finding. The assumption has always been that recently evolved proteins should play relatively minor (and/or redundant) roles in basic processes such as development and organogenesis. Similar arguments could be made about TEs, although the paradigm is starting to shift on this matter. This paper certainly contributes to that shift, uncovering a firm example of how the KZPF-TE relationship has shaped human evolution.

1. The study is generally well conducted, with solid genetic data and extensive analysis of the in vitro KO model. My main comment for improvement is related to the fact that all hESC results appear to have been derived from a single clone. Although the data presented from this clone are convincing, it would be reassuring to see key results replicated in multiple null CRISPR clones (with an equal number of wildtype clones derived during CRISPR). I am assuming here that obtaining a null clone wasn't particularly difficult, and that multiple null clones were stored, which might not be the case.

Alternatively, or complementarily, if patient-derived iPSCs were available, this would also add robustness to the study.

We agree that replicating the key results in an independent clone would support and validate our model and our findings. To address the reviewer's comment, we present data from an iPSC line derived from patient 2 who is homozygous for a deletion of exons 4 and 5 of ZNF808 which we differentiated up to the posterior foregut stage (S3). This approach has the advantage of not requiring CRISPR to introduce the genetic defect but as the iPSC was derived from a patient with a homozygous ZNF808 deletion, we cannot correct the pathogenic variant and hence create an isogenic control.

Nonetheless, we find excellent agreement between our ZNF808 KO and the iPSC line in terms of loss of H3K9me3, gain of H3K27ac, and changes in gene expression (shown in panels a and b below). However, despite the excellent agreement, the clone was difficult to obtain and work with. For this revision, we have now succeeded in obtaining a new iPSC clone from the same patient. This clone performs better, and we have performed qPCR on selected liver-associated genes at the S3 stage to further validate our results – results agree with our ZNF808 KO H1 cells (panel c below).

These results could be included in the paper if the reviewers consider it appropriate.

Figure legend – Validation of epigenetic and transcriptomic dysregulation observed in the *ZNF808* KO cells using patient-derived iPSCs.

- a. Heatmap of the 220 MER11 elements that lose H3K9me3 and gain H3K27ac in the *ZNF808* KO as presented in figure 3, with the addition of signal obtained when differentiating iPSCs up to S3.
- b. Left, boxplots showing fold change of activated and repressed dysregulated genes identified in *ZNF808* KO and patient-derived iPSCs, showing agreement in direction and magnitude of the gene expression perturbation. Boxes mark interquartile range, with central line describing the median and whiskers 1.5x interquartile range. Right, linear regressions between *ZNF808* KO and iPSC log2 fold change KO over WT at same set of genes. Regression coefficients and p-value of slope term given.
- c. qRT-PCR for five hepatic marker genes assayed at the posterior foregut stage (S3) in cells derived from H1 control, H1-*ZNF808*-KO and the patient iPSC (line HEL340.7) carrying the *ZNF808* deletion (n = 3-4 independent differentiation experiments. Data represents mean \pm SEM. t test *p<0.05, **p<0.01).

2. My only other comment is minor: it would be useful to provide the number of elements of each MER11 subfamily within each of the clusters in Figure 3c. Mainly because I think this will point to a major role of MER11A in driving liver-specific gene expression. This is the subfamily that bears the relevant motifs and binds HNF4A, for example. Also, in Supp. Fig. 8d, virtually every example of hepatocyte-active elements comes from the MER11A subfamily.

We have performed this analysis which is now provided as part of an expanded Supplementary Figure 6. Each cluster contains a mix of MER11A, B and C – some trends are visible, as noted by the reviewer, with more elements of the MER11A family in the clusters that gain H3K27ac activity in the ZNF808 KO at late stages of differentiation (#1, #4 and #5), which fits well with our hypothesis that late clusters of active MER11 elements are bound by HNF4a and members of the GATA family. However, the numbers are very small and at this stage we do not want to draw any conclusions nor speculate on these trends in the main manuscript.

Miguel Branco

Reviewer #3:

Remarks to the Author:

Summary of the key results

The manuscript “Primate-specific ZNF808 is essential for pancreatic development in humans” by De Franco et al. identifies a novel source for congenital pancreas agenesis – mutation in the DNA binding gene ZNF808. Notably, this gene is primate-specific, and as such this is “the first report of loss of a primate-specific gene causing a congenital developmental disease”. The authors move on to uncover the mechanism underlying the pancreas’s lack of ability to form, and place the finger on ZNF808’s role as a suppressor of the MER11 family of transposable elements. In its absence, the elements that are normally silenced become activated, and their transcription dysregulates the balance between the pancreatic and the hepatic gene expression programs.

The identification of a novel mutation responsible for pancreas agenesis is very interesting, and it being

in a gene unique to primates further increases the interest of the work. In addition, the gene's mechanism of action – as presented by the authors – is fascinating. However, as detailed below, I believe that several important claims made by the authors are not adequately supported by their findings. Briefly, the manuscript can be divided into two parts. In the first, the authors nicely show that mutations in ZNF808 can be linked to pancreas agenesis. This claim is supported by highly convincing data. In the second part of the manuscript, the authors describe ZNF808's mechanism of action. The evidence in this part is lacking, and in fact brings to questions why it's known effect is limited to the pancreas.

Major concerns

1) To prove ZNF808's mechanism of action, the authors use an in vitro protocol for the directed differentiation of pluripotent cells into pancreatic progenitors, and observe an impact of ZNF808 KO on the cells' epigenetic and transcriptional landscape:

1a. It is unclear if the authors see any effect of the gene's deletion on the quality of pancreas differentiation. If no effect is observed, the relevance of the in vitro model to the issue at hand can be questioned. Perhaps examining the effect of the gene's loss on later stages of differentiation (e.g. endocrine differentiation) may reveal a stronger phenotype.

In our model, the strongest effect observed in the ZNF808-KO was the upregulation of the hepatic program rather than impairment of pancreatic differentiation. Specifically, there was no obvious impairment in in vitro differentiation to beta-like cells. This result is not unexpected since our model is based on directed, not spontaneous differentiation.

Current differentiation protocols, including the one used in this study, have been extensively optimized to yield abundant numbers of pancreatic progenitor cells from pluripotent stem cells in a few days. To achieve this, the differentiation media contain potent small molecule inhibitors and growth factors that force the differentiation of endodermal cells into the pancreatic lineage and inhibit liver fate (during S3 and S4 retinoic acid and Sonic Hedgehog signalling inhibitor SANT1 are used to promote PDX1 expression, FGF7 promotes pancreatic endoderm proliferation, while BMP signalling inhibitor LDN-193189 promotes pancreas and inhibits liver specification). These differentiation protocols have been previously reported to override genetic defects leading to neonatal diabetes with or without pancreatic agenesis (such as variants in GATA6, PTF1A and MNX1 (PMID: 28196690, 27133796)). We observed the same in our model, with no obvious defect in beta cell differentiation observed when carrying the differentiation beyond pancreatic progenitor stage (S4). However, even when using this pancreatic-directed protocol, we observe a strong upregulation of liver genes in the ZNF808-KO cells at S2, S3 and S4 stages, indicating that these cells are more prone to differentiate into the alternative lineage.

To further examine the propensity of ZNF808-KO cells to acquire a liver identity, we have now conducted a new set of differentiation experiments without the inhibitors and growth factors that induce the pancreatic fate (NO FACTORS condition for 4 days at the posterior foregut stage, see figure below) and

compared it to the standard differentiation protocol (STANDARD condition for 4 days). The results indicate that ZNF808-KO cells have a higher propensity than wild type control to differentiate into the liver lineage when the pancreas inducing signalling cues (retinoic acid, FGF7, Sonic Hedgehog and BMP signalling inhibition) are omitted from the media, as shown by the increased expression of liver marker genes including MER11-proximal genes (BAAT, TDO2). However, in these experimental conditions it is not possible to continue differentiation toward pancreas.

qRT-PCR of posterior foregut cells (Stage 3, S3) derived from H1 control and H1-ZNF808-KO stem cells with the standard pancreatic-directed differentiation protocol for 4 days (STANDARD) or without pancreas inducing cues for 4 days (NO FACTORS) (n = 3-4 independent differentiation experiments. Data represents mean \pm SEM. t test *p<0.05, **p<0.01)

1b. Pancreas agenesis includes malformation of both the acinar and the endocrine components of the pancreas, but the authors do not test a protocol for acinar differentiation. This may be excluded if the endocrine differentiation used would have shown a substantial phenotype.

The reviewer has raised a very good point which would be extremely interesting to explore in the future. However, we feel that this would be beyond the scope of the current short communication which is focused on the genetic findings and insights in ZNF808's function during early human development.

1c. The most significant effect of ZNF808 KO on MER11 expression seems to be in S0, when cells are still in their undifferentiated state. However, there is no mention of a potential effect on the differentiation into tissues other than the pancreas – preferably non-endodermal tissues - to examine if this KO indeed affects only the pancreas.

We thank the reviewer for raising this interesting point. Our rationale for focusing on the pancreas in our letter was driven by the phenotype of patients with homozygous loss-of-function variants in ZNF808. Despite the effect on MER11 activation at S0 and the widely detected expression of ZNF808 in embryonic and adult tissues, the patients' phenotype showed that only pancreas development was affected, with no additional extra-pancreatic features.

Additional evidence supporting the tissue-specific effect of ZNF808 loss comes from our gene set enrichments presented in Supplementary Figure 9d and Supplementary Table 5. In this analysis we did not identify any coherent gene expression programme in genes dysregulated at S0 that would implicate any other lineage than the skew toward liver cell fate we focussed on.

Prompted by the reviewer's comment, we have further investigated the possible impact of ZNF808-KO on the potential of stem cells to differentiate into other lineages by conducting spontaneous differentiation experiments using the embryoid body approach with ZNF808 KO and control H1 stem cells. The results indicate that the ZNF808 KO cell lines can give rise to cell types from the three germ layers. We could include these new results in a supplementary figure if that is considered appropriate.

Immunofluorescence stainings for ectoderm (tubulin IIIb, TUBB3), mesoderm (smooth muscle actin, SMA) and endoderm (SOX17) markers of spontaneously differentiated embryoid bodies derived from control H1 and ZNF808-KO stem cells demonstrating their pluripotentiality to differentiate into the three germ layers. Scale bar = 200 micrometers.

2) Based on transcriptional analysis, the authors claim that “the unmasking of MER11 elements in the ZNF808 KO has direct impact on gene expression”, and present this impact as the underlying mechanism of ZNF808's mode of action. However, the manuscript does not show any direct evidence for an effect of MER11's transcription on gene expression. The data only shows correlation between the effect of ZNF808's KO on MER11's expression and on the expression of the dysregulated genes. It could be, for example, that ZNF808 is directly responsible for both the gene's change in expression and, independently, on MER11 expression.

We understand the reviewer's concerns, we have amended the text to remove the word “direct”.

However, we do believe that direct ZNF808 action via MER11 is the most parsimonious and compelling interpretation of the data. We have conducted additional analysis to address the reviewer's concerns. In the ZNF808-KO MER11 elements become active, acting as enhancers at which transcription factors bind

resulting in the activation of nearby genes. We demonstrate that there is no relationship between the activated genes and ZNF808 binding sites outside the MER11 elements:

Figure legend $-\log_{10}$ Fisher exact test p-values for the association between genes activated or repressed and genes lying within the given distance of ZNF808 binding sites either on or outside the 220 unmasked MER11 elements.

ZNF808 binding outside the 220 MER11 elements cannot explain any gene expression changes. If ZNF808 were acting directly on these genes through binding outside MER11 elements, we would expect to see a correlation, but we do not. We are yet to confirm precisely which genes are directly regulated from which MER11 elements that trigger the liver-associated genetic programme, but from this analysis we are confident MER11 elements drive a subset of early gene activation that have downstream consequences.

Additional issues

3) supplementary figure 6 and Figure 2c are difficult to interpret.

We have improved figure 2c to help interpretation – we have reversed the orientation of the figure so that younger elements are on the right instead of the left, added a more informative title and legends, added markings for the highest number of elements for each subfamily, and increased visibility of markers and legends. We hope that this resolves the issue, and we are open to further suggestion on how we could improve this showcase of the evolutionary dynamics of MER11 elements in the human genome.

Supplementary figure 6 is now expanded to address comments by other reviewers – panel c was the only one present in the original version and this has been improved with an informative title and colour bar. We also provide the same panel shown in figure 3 illustrating the clusters of activated MER11 elements for reference, displayed in panel a of this expanded supplementary figure.

4) In supplementary Figure 1b – in situ hybridization of ZNF808 during pancreas development – the

staining is not convincing. It is hard to see in which cells it is expressed. In addition, a non-pancreatic control sample could be helpful, to show that staining is indeed specific.

We thank the reviewer for the comment. Subtle transcript levels can be difficult to characterise by in situ hybridisation, particularly in ubiquitously expressed genes such as ZNF808. From the images we agree with the reviewer that it is difficult to see in which cells it is expressed (in situ hybridisation offering less dynamic range than some other staining methods); however, what is clear is there is heterogeneity in gene expression within the pancreas at each time point.

As highlighted throughout the manuscript, the expression of ZNF808 is detected at various levels in different tissues during development and is thus not exclusive to the pancreas. The images are presented to show its spatial expression during development. What can be appreciated are relative differences in ZNF808 transcript expression between tissues; as an example, there is less transcript detection in the surrounding mesenchyme. The RNA-seq data we also show in the same supplemental figure provides a better quantitative picture of the variation in expression between tissues.

5) In figure 1b – can a statistical value be assigned to the likelihood of the findings? i.e. what is the statistical significance of the observed pattern of inheritance?

The aim of Figure 1b is to show the patients' pedigrees and co-segregation of the variants. While common in gene discovery papers, this kind of display item does not usually include statistics. Whilst not directly linked to this figure, we can estimate the statistical significance of our findings based on the absence of ZNF808 deleterious homozygous loss of function variant in > 680,000 individuals without pancreatic agenesis (UK BioBank (n=454,756), Gnomadv2.1.1 (n=141,071), 100,000 genomes project from Genomics England (n=75,118), Genes and Health (n=8,921) and 875 congenital hyperinsulinism probands). The possibility of such results occurring by chance is $p < 1 \times 10^{-38}$ (Fisher's Exact Test).

Decision Letter, first revision:

27th Jun 2023

Dear Professor Hattersley,

Your Letter, "Primate-specific ZNF808 is essential for pancreatic development in humans" has now been seen by your 3 original referees. We are interested in the possibility of publishing your study in Nature Genetics, but would like to consider your response to the remaining concerns of Reviewer #3, and the remaining request from Reviewer #2, in the form of a revised manuscript before we make a final decision on publication.

We therefore invite you to revise your manuscript. Please highlight all changes in the manuscript text file. At this stage we will need you to upload a copy of the manuscript in MS Word .docx or similar

editable format.

*2) If you have not done so already please begin to revise your manuscript so that it conforms to our Letter format instructions, available [here](http://www.nature.com/ng/authors/article_types/index.html). Refer also to any guidelines provided in this letter.

[redacted]

We hope to receive your revised manuscript within four to eight weeks. If you cannot send it within this time, please let us know.

Sincerely,

Safia Danovi
Editor
Nature Genetics

Reviewers' Comments:

Reviewer #1:

Remarks to the Author:

The authors went the distance in answering all our previous concerns. This is a very interesting story, placing an under-explored family of proteins in the limelight. Well done!

Reviewer #2:

Remarks to the Author:

The authors have addressed my comments, although data for my first point on validation in a different hESC/iPSC clone was only included in the the rebuttal letter. I appreciate the authors' reservations about the new iPSC data not having an appropriate isogenic control, but in my opinion this still goes a long way in providing reassurances that the effects are reproducible. My personal view is that it should be included in the manuscript, mentioning the control caveat.

Miguel Branco

Reviewer #3:

Remarks to the Author:

De Franco et al. have addressed most of the issues raised, and their arguments now seem more compelling. I would recommend publishing the paper under the following reservations:

- a. The authors should explicitly state in the discussion section that the mechanism they offer (Mer11 masking) is parsimonious, but not conclusive. They should suggest an experiment that proves it beyond circumstantial relationship (e.g. KO of Mer elements if possible). The conduct of such an experiment SHOULD NOT be included in this publication.
- b. The authors chose to wave the comment regarding the quality of in situ hybridization in supplementary figure 1b, and they claim that it clearly shows heterogeneity in expression within the pancreas. I do not see such heterogeneity, and I doubt if what I see is staining for ZNF808. As

requested, the authors should produce an image from a tissue that does not express ZNF808, to prove that their staining is correct.

Author Rebuttal, first revision:

Rebuttal to Reviewers' Comments:

Reviewer #1:

Remarks to the Author:

The authors went the distance in answering all our previous concerns. This is a very interesting story, placing an under-explored family of proteins in the limelight. Well done!

We are pleased the reviewer was satisfied with our responses.

Reviewer #2:

Remarks to the Author:

The authors have addressed my comments, although data for my first point on validation in a different hESC/iPSC clone was only included in the the rebuttal letter. I appreciate the authors' reservations about the new iPSC data not having an appropriate isogenic control, but in my opinion, this still goes a long way in providing reassurances that the effects are reproducible. My personal view is that it should be included in the manuscript, mentioning the control caveat.

Miguel Branco

We are happy to include the iPSC data showing validation of our findings in the manuscript. We have added the iPSC data to the supplementary appendix (Supplementary figure 11 in Supplementary Figures_R2) detailing that there is not an isogenic control in the figure legend and added the details of the methodology to Supplementary Information_R2. We have referenced this information in the main text.

Reviewer #3:

Remarks to the Author:

De Franco et al. have addressed most of the issues raised, and their arguments now seem more compelling. I would recommend publishing the paper under the following reservations:

a. The authors should explicitly state in the discussion section that the mechanism they offer (Mer11 masking) is parsimonious, but not conclusive. They should suggest an experiment that proves it beyond circumstantial relationship (e.g. KO of Mer elements if possible). The conduction of such an experiment SHOULD NOT be included in this publication.

We have now included a statement in the discussion to address the reviewer's comment. The end of the second paragraph in the discussion now reads: 'We provide correlative evidence that the unmasking of

MER11 elements in the ZNF808 KO ultimately leads to the downstream induction of a liver gene expression programme during pancreas differentiation. Future work combining targeted genome editing and 3D conformation data will be necessary to confirm which of the regulatory regions repressed by ZNF808 are involved in pancreas development.'

b. The authors chose to wave the comment regarding the quality of in situ hybridization in supplementary figure 1b, and they claim that it clearly shows heterogeneity in expression within the pancreas. I do not see such heterogeneity, and I doubt if what I see is staining for ZNF808. As requested, the authors should produce an image from a tissue that does not express ZNF808, to prove that their staining is correct.

ZNF808 is broadly expressed in every human adult and fetal tissue tested (as now shown in Supplementary Figures 1a and 1b). This means that we are unable to provide in situ hybridization for a tissue that does not express ZNF808 to give the requested proof that the staining is specific. We have therefore decided to remove the in-situ hybridisation data from the manuscript.

We believe these results obtained using the BaseScope methodology with appropriately specific probe sequence and standard positive and negative controls is accurate, however we are also convinced that removing that figure does not impact our overall message or any conclusion. The RNA-seq data provided in Supplementary Figure 1b is a more reliable and quantitative measure of ZNF808 expression and this shows that it is highly expressed in the fetal pancreas.

We have now added a new Supplementary Figure 1a showing ZNF808 expression across adult tissues highlighting its broad expression in adult, to supplement the existing panel 1b of fetal tissues. The main manuscript has been updated to reflect these changes.

Decision Letter, second revision:

11th Jul 2023

Dear Dr. Hattersley,

Thank you for submitting your revised manuscript "Primate-specific ZNF808 is essential for pancreatic development in humans" (NG-LE61919R1). It has now been seen by the original referees and their comments are below. The reviewers find that the paper has improved in revision, and therefore we'll be happy in principle to publish it in Nature Genetics, pending minor revisions to satisfy the referees' final requests and to comply with our editorial and formatting guidelines.

Sincerely,

Safia Danovi
Editor
Nature Genetics

Final Decision Letter:

10th Oct 2023

Dear Dr Hattersley,

I am delighted to say that your manuscript "Primate-specific ZNF808 is essential for pancreatic development in humans" has been accepted for publication in an upcoming issue of Nature Genetics.

Your paper will be published online after we receive your corrections and will appear in print in the next available issue. You can find out your date of online publication by contacting the Nature Press Office (press@nature.com) after sending your e-proof corrections. Now is the time to inform your Public Relations or Press Office about your paper, as they might be interested in promoting its publication. This will allow them time to prepare an accurate and satisfactory press release. Include your manuscript tracking number (NG-LE61919R2) and the name of the journal, which they will need

when they contact our Press Office.

Please note that *Nature Genetics* is a Transformative Journal (TJ). Authors may publish their research with us through the traditional subscription access route or make their paper immediately open access through payment of an article-processing charge (APC). Authors will not be required to make a final decision about access to their article until it has been accepted. [Find out more about Transformative Journals](https://www.springernature.com/gp/open-research/transformative-journals)

Authors may need to take specific actions to achieve [compliance](https://www.springernature.com/gp/open-research/funding/policy-compliance-faqs) with funder and institutional open access mandates. If your research is supported by a funder that requires immediate open access (e.g. according to [Plan S principles](https://www.springernature.com/gp/open-research/plan-s-compliance)) then you should select the gold OA route, and we will direct you to the compliant route where possible. For authors selecting the subscription publication route, the journal's standard licensing terms will need to be accepted, including [self-archiving-and-license-to-publish](https://www.nature.com/nature-portfolio/editorial-policies/self-archiving-and-license-to-publish). Those licensing terms will supersede any other terms that the author or any third party may assert apply to any version of the manuscript.

If you have not already done so, we invite you to upload the step-by-step protocols used in this manuscript to the Protocols Exchange, part of our on-line web resource, natureprotocols.com. If you complete the upload by the time you receive your manuscript proofs, we can insert links in your article that lead directly to the protocol details. Your protocol will be made freely available upon publication of your paper. By participating in natureprotocols.com, you are enabling researchers to more readily reproduce or adapt the methodology you use. [Natureprotocols.com](https://natureprotocols.com) is fully searchable, providing your protocols and paper with increased utility and visibility. Please submit your protocol to <https://protocolexchange.researchsquare.com/>. After entering your [nature.com](https://www.nature.com) username and password you will need to enter your manuscript number (NG-LE61919R2). Further information can be found at <https://www.nature.com/nature-portfolio/editorial-policies/reporting-standards#protocols>

Sincerely,

Safia Danovi
Editor
Nature Genetics